# Towards understanding the pattern of glacier mass balances in High Mountain Asia using regional climatic modelling

Remco J. de Kok[1], Philip D.A. Kraaijenbrink[1], Obbe A. Tuinenburg[2], Pleun N.J. Bonekamp[1], Walter W. Immerzeel[1]

[1]Department of Physical Geography, Utrecht University, Utrecht, PO Box 80115, 3508 TC, The Netherlands
[2]Copernicus Institute of Sustainable Development, Utrecht University, Utrecht, PO Box 80115, 3508 TC, The Netherlands

*Correspondence to*: Remco J. de Kok (r.j.dekok@uu.nl)

**Abstract.** Glaciers in High Mountain Asia provide an important water resource for communities downstream and they are markedly impacted by global warming, yet there is a lack in understanding of the observed glacier mass balances and their spatial variability. In particular, the glaciers in the western Kunlun Shan and Karakoram ranges (WKSK) show neutral to positive mass balances despite global warming. Using models of the regional climate and glacier mass balance, we reproduce the observed patterns of glacier mass balance in High Mountain Asia of the last decades within uncertainties. We show that low temperature sensitivities of glaciers and an increase in snowfall, for a large part caused by increases in evapotranspiration from irrigated agriculture, result in positive mass balances in WKSK. The pattern of mass balances in High Mountain Asia can thus be understood from the combination of changes in climatic forcing and glacier properties, with an important role for irrigated agriculture.

## 1. Introduction

Glaciers in High Mountain Asia (HMA, see Fig. 1) show a very diverse response to the changing climate. Most glaciers are losing mass, as expected, but surprisingly, glaciers are stable or growing in a region northwest of the Tibetan plateau, a phenomenon dubbed the Karakoram anomaly (Hewitt, 2005). Recent studies have mapped glacier mass losses and velocity changes in detail and have shown that the regions of largest glacier growth and acceleration are the Kunlun Shan and parts of the Pamir, with the glaciers in Karakoram being close to a steady state (Brun et al., 2017; Dehecq et al., 2019; Gardelle et al., 2012, 2013; Kääb et al., 2015; Lin et al., 2017). Part of the mass balance variability seems correlated to differences in the temperature sensitivity, i.e. the change of mass balance for a certain change in temperature, of the glaciers (Sakai and Fujita, 2017), but this alone cannot explain why some glaciers are actually growing, since either a decrease of ablation or an increase in accumulation is needed.

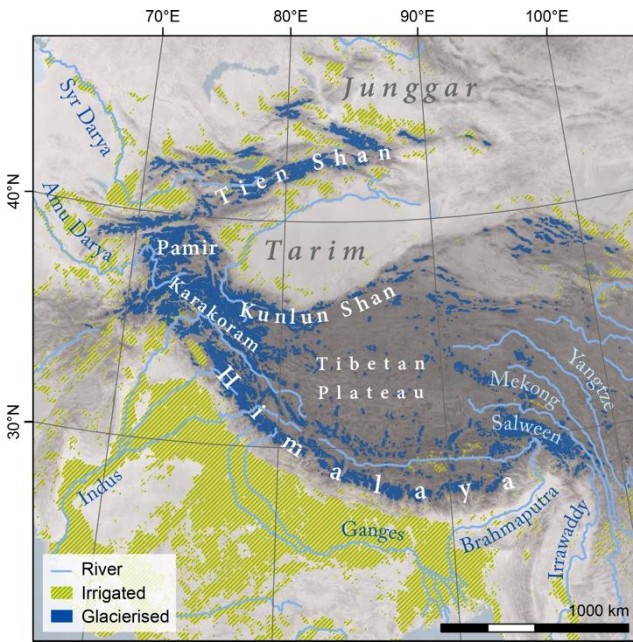

**Figure 1: Map of study area. Irrigated areas (from GMIA (Siebert et al., 2010)) and glacierised areas (from RGI 6.0 (Pfeffer et al., 2014)) are indicated.**

Suggested causes of the Karakoram anomaly include an increase in winter snowfall (Cannon et al., 2015; Kapnick et al., 2014; Norris et al., 2015, 2018), summertime cooling (Bocchiola and Diolaiuti, 2013; Forsythe et al., 2017; Fowler and Archer, 2006; Khattak et al., 2011; Ul Hasson et al., 2017), and an increase in summertime precipitation and clouds due to irrigation in the agricultural regions adjacent to the Kunlun Shan and Pamir (de Kok et al., 2018). So far, these hypotheses have only tried to explain the Karakoram anomaly in qualitative terms, identifying possible climatic conditions that could lead to glacier growth. None of these have yet been able to directly reproduce the observed pattern of glacier mass balances in HMA directly by showing the response of the glaciers to the historic climatic forcing. Here, we simulate glacier mass balances using modelled time series of temperature and snowfall from a regional climate model to reproduce the pattern of observed mass balances in HMA, and to more deeply understand the underlying causes.

## 2. Methods

### 2.1 Regional climate model

Irrigation can influence the regional and global climate (Cook et al., 2015; Lee et al., 2011; Lobell et al., 2008; Puma and Cook, 2010; Sacks et al., 2009). Since the regions surrounding HMA host the largest irrigated areas in the world, e.g. the Indo-Gangetic Plain (see Fig. 1), irrigation potentially influences the regional climate in HMA (Cai et al., 2019; de Kok et al., 2018).

However, available re-analysis datasets do not fully include irrigation, and generally have a relatively coarse spatial resolution. Hence, we downscaled ERA-Interim (Dee et al., 2011) re-analysis data using the Weather Research and Forecasting model (WRF, Skamarock & Klemp, 2008) to obtain a climate dataset between 1980-2010 for High Mountain Asia with a resolution higher than that of ERA-Interim. We artificially applied irrigation to the surface by adding a precipitation term each time step, with a rate that is determined per month. The precipitation is added to the NOAH-MP surface module, but the atmospheric module is not altered. The amount of irrigation applied per grid cell was based on the monthly water demand, which indicates the amount of irrigation needed to compensate evapotranspiration, after subtraction of the precipitation, that was calculated by the PCRaster Global Water Balance model (PCR-GLOBWB; van Beek & Bierkens, 2008; van Beek, Wada, & Bierkens, 2011; Van der Esch et al., 2017; Y. Wada, Wisser, & Bierkens, 2014; Yoshihide Wada et al., 2011). In this way, the irrigation amount is not easily overestimated, as could be the case when e.g. soil moisture would be kept constant. In reality, insufficient water might be available to meet the predicted demand, whereas inefficient irrigation will result in a larger water gift than predicted. The PCR-GLOBWB runs were forced by WATCH data, based on ERA-Interim (Weedon et al., 2014).

We used WRF, version 3.8.1, to downscale ERA-Interim data to a resolution of 20x20 km, with 50 vertical levels. Settings are nearly identical to our previous work (de Kok et al., 2018), which are based on the work of Collier and Immerzeel (2015) and Bonekamp et al. (2018) and are shown in Table 1. Additionally, we now use grid nudging of the upper 35 vertical levels for horizontal wind, temperature, and humidity, as opposed to only forcing the model at the boundary in our previous study. This ensures the large-scale upper-atmospheric circulation closely follows that of ERA-Interim, whereas near the surface, the model is more determined by the physics in WRF, e.g. evaporation in irrigated areas. The nudged levels and the values of the nudging parameters have been found to perform well in similar studies (Collier and Immerzeel, 2015; Otte et al., 2012), and are: 0.0001, 0.0001, and 0.00005 $s^{-1}$ for horizontal winds, temperature and water vapour, respectively. The default values for all three parameters are 0.0003 $s^{-1}$ in WRF.

**Table 1: Physics modules and assumptions used in WRF.**

| Module | Setting |
|---|---|
| Radiation | RRTMG scheme (Iacono et al., 2008) |
| Microphysics | Morrison scheme (Morrison et al., 2009) |
| Cumulus | Kain-Fritsch (new Eta) scheme (Kain, 2004) |
| Planetary boundary layer | YSU scheme (Hong et al., 2006) |
| Atmospheric surface layer | MM5 Monin-Obukhov scheme (Beljaars, 1995; Dyer and Hicks, 1970; Paulson, 1970; Webb, 1970; Zhang and Anthes, 1982) |
| Land surface | Noah-MP (Niu et al., 2011) |

| Top boundary condition | Rayleigh damping |
| --- | --- |
| Diffusion | Calculated in physical space |
| Nudging | Grid-point *u, v, T, q* above level 15 |

Annual concentrations of $CO_2$, $CH_4$, and $N_2O$, which are manually set in the RRTMG radiation module, are derived from NOAA (Dlugokencky et al., 2018) and AGAGE (Prinn et al., 2000) data, as aggregated by the European Environment Agency
(www.eea.europa.eu, accessed March 2018), and are kept constant throughout each year. We used a 10-day spin-up for each month and run each month separately to be able to include a different irrigation amount each month. Monthly initialisation of the snow cover, surface and soil temperature, and surface moisture was taken from GLDAS 2.0 (Rodell et al., 2004) monthly mean values. We checked whether temperatures and precipitation at the end of a month agreed with those at the end of the spin-up period for the subsequent month and they agreed within a few percent for all selected points. Results are output every
6 hours.

**2.2 Glacier model**

To assess the response of the glaciers to the atmospheric forcing, we employ a glacier mass balance gradient model (Kraaijenbrink et al., 2017). The model assumes a calibrated mass balance gradient along the glacier, and parameterises downslope mass flux in a lumped procedure that is based on vertical integration of Glen's flow law (Marshall et al., 2011). It
also includes a parameterisation for the effects of supraglacial debris on surface mass balance (Kraaijenbrink et al., 2017), i.e. enhancing melt in the case of a shallow debris layer and limiting melt for thicker debris (östrem, 1959). We modelled all individual glaciers in HMA larger than 0.4 $km^2$ (n=33,587) transiently for the period 1980-2010 (Kraaijenbrink et al., 2017). For ease of comparison with published observations, we select only those larger than 2 $km^2$ for the final analysis, which represent 95% of the glacier volume in HMA. Initial mass balance conditions in 1980 were set to be stable, while all other
initial and reference conditions as described in the original study (Kraaijenbrink et al., 2017) were maintained. That is, using ERA-Interim data to locally calibrate the mass balance gradient of each glacier by constraining maximum ablation by a downscaled positive degree day climatology at the glacier terminus, and maximum accumulation by mean annual precipitation over the entire glacier area. The model simulates glacier mass change and evolution using a one-year time step, and hence requires representative annual input of temperature and precipitation. These are used to shift the mass balance curve according
to sensitivity of the glacier's equilibrium line altitude to temperature changes, and adapt the maximum accumulation according to changes in precipitation (Kraaijenbrink et al., 2017). The aforementioned model is relatively simple, but such simplicity is required to model the thousands of glaciers in the region within a reasonable time. Other models that aim to model glacier mass balances over such a large scale are also relatively simple, and the output of our model is close to the median of similar models for High Mountain Asia, with our model being the only one that treats debris cover (Marzeion et al., 2020).

To modulate the curve transiently, we applied annual precipitation changes derived from annual changes in WRF snowfall and air temperature changes determined from annual changes in WRF melt season temperatures, i.e. when average daily air temperature is above -5 °C. A threshold value of 0 °C did not significantly change the glacier mass balance results for most of HMA, but meant that temperature changes for the coldest points could not be determined, since they are always below 0 °C. We did not take into account whether the WRF grid point was glacierised or not. To reduce potential biases imposed by spurious reference conditions, the reference for the changes in air temperature and precipitation was taken to be the average of the first six modelling years, i.e. 1980-1985. We performed three separate glacier model runs to evaluate the attribution of snowfall and temperature to the glacier mass changes in our model, forced by: (1) precipitation and air temperature, (2) only air temperature, and (3) only snowfall. To illustrate the temperature and precipitation sensitivity of the glaciers, we also performed calculations using a fixed air temperature or snowfall trend for all of HMA, with the other variable kept constant.

## 2.3 Moisture tracking

Our moisture tracking model (Tuinenburg et al., 2012) follows the moisture associated with precipitation backwards in time to determine where the moisture first enters the atmosphere. It therefore establishes a direct causal link between evapotranspiration and precipitation downwind. For the moisture tracking, we clustered locations that have similar climates in terms of seasonality, since these will likely also have similar moisture sources. For the clustering, we used the monthly mean values of precipitation, horizontal wind fields at 400 hPa, and 2m-temperatures, with means subtracted and divided by the standard deviations, to perform a k-means clustering using 13 clusters. In this way, we delineate regions that have similar surface variables, relevant for the glaciers. Furthermore, these regions are also under the influence of similar winds, relevant for the moisture transport. We performed the clustering with different numbers of clusters and found 13 to give reasonably-sized, yet distinct areas, while also being close to the knee in the total distance away from the cluster means. We perform the tracking on two of these clusters, indicated later in Fig. 14, which are close geographically, but have contrasting snowfall trends.

We perform the moisture tracking by releasing moisture parcels from the target area at random positions within the area and advecting them backwards in time using interpolated wind fields on fixed pressure levels. The amount of evaporation $A$ (mm) that contributes to the precipitation in the target area, at a given location $x,y$ and time step $t$, depends on the evapotranspiration ET (mm), the total amount of all water in the parcel $W_{parcel}$ (mm), the fraction of water in the parcel that evaporated from the source $S_{target}$, and the total precipitable water in the column TPW (mm):

$$A_{x,y,t} = \mathrm{ET}_{x,y,t} \frac{W_{parcel,t} S_{target,t}}{\mathrm{TPW}_{x,y,t}} \quad , \tag{1}$$

The amount of water in the parcel is then updated every time step, including the precipitation $P$ that adds to the parcel when moving back in time.


$$W_{parcel,t-1} = W_{parcel,t} + \left(P_{x,y,t-1} - \text{ET}_{x,y,t-1}\right)\frac{W_{parcel,t}}{TPW_{x,y,t-1}} \qquad (2)$$

The fraction **of** precipitation in the target area that originates from a certain source area is then updated as follows:

$$S_{target,t-1} = \frac{S_{target,t}W_{parcel,t} - A_{x,y,t-1}}{W_{parcel,t-1}} \qquad (3)$$

We track the parcel until either more than 99% of the target precipitation is tracked to a source area, or the tracking time is more than 30 days.

Within the WRF domain, the parcels are advected and the moisture budget is calculated using the WRF wind fields and water fluxes. When a parcel gets within one degree of the edge of the WRF domain, there is a gradual linear change to a use of ERA-Interim reanalysis data to ensure continuous movement of the air parcels over the domain edge. Within one degree distance from the domain edge, the values used to do the moisture tracking are a combination of the WRF and ERA-Interim values: $y_{int}$ = $d*y_{WRF}$ + $(1-d)*y_{ERA}$ , where $d$ is the distance to the edge. Outside of the WRF domain, the ERA-Interim values are used.


We noted that the surface moisture flux in ERA-Interim is on average 50% higher than in WRF, resulting in a higher mean and standard deviation of the moisture sources outside the WRF domain. Unfortunately, this systematic offset between the two datasets cannot be easily remediated. Although this will not change the spatial patterns of the moisture source trends in a major way, the absolute values of the trends will be lower in the WRF domain than outside if there is a scaling factor in moisture

flux between the two datasets. The trends in the Tarim basin will then be underestimated with respect to regions such as the Caspian Sea and the Junggar basin.

### 2.4 Statistics

Pearson correlation coefficients are calculated between pairs of different datasets (e.g. Figs. 2-5) using the vectors of annual

or seasonal mean values, with one value for each year. The figures indicate over which period the mean is taken for each year. The trends shown in Figs. 2-5, 8, and 17 are the slopes from linear fits to these vectors. P-values for the correlations are determined using the beta function, as implemented in SciPy (Virtanen et al., 2020).

## 3. Results

### 3.1 Validation and comparison

Any attempt to understand the Karakoram anomaly is greatly hampered by the almost complete lack of *in situ* meteorological data in WKSK. The sparse weather stations in the region are often situated at relatively low elevation, or in urban environments, and poorly represent the high mountain climate. Furthermore, different types of precipitation datasets seem to greatly underestimate the precipitation in mountainous terrain (Immerzeel et al., 2015; Ménégoz et al., 2013; Palazzi et al., 2013). These complications imply that any meteorological dataset, including reanalysis datasets, are associated with relatively large

fundamental errors in WKSK, which prevents reliable validation of any model of WKSK, such as the one presented in here. Although not covering the glacierised areas of interest, we compared our WRF output with data of the region surrounding WKSK, to ensure that the WRF output is a reasonable representation of the regional climate between 1980-2010. Since the glacier model requires annual input, representation of the interannual variability is especially important. Any constant biases are of less importance, since we use relative interannual variations as input for the glacier model. However, biases in

temperature will have an effect on the snow-rain partition.

We collected meteorological station data from the Global Historical Climatology Network (GHCN, Lawrimore et al., 2011, accessed June 2019), and selected those that have at least 15 years of full data between 1980-2010. To be able to compare the WRF output with the station data, we apply a simple downscaling to the WRF temperatures in the grid that includes the station.

We fit a linear temperature lapse rate to the temperatures and grid altitudes of a 2x2° box surrounding the station location. We then correct the WRF temperature by applying the lapse rate to the difference in altitude between the WRF grid and the station. Precipitation can also change significantly with location, but there is no clear relation between precipitation and altitude (Bonekamp et al., 2019; Collier and Immerzeel, 2015). For this simple comparison, we do not apply a downscaling of the WRF precipitation.


Our WRF output produces May-September temperatures that are generally higher than the stations in the Tarim basin. However, biases are generally very low on the Tibetan Plateau, with values around 1°C (Fig. 2a). The median root-mean-square deviation between WRF and the stations for the time-series of seasonal mean temperatures is 1.8°C. The stations generally indicate a strong heating trend (Fig. 2b), but also show relatively large differences for close-by stations. Correlations

between the annual variations in annual mean temperatures and mean temperatures between May-September are given in Fig. 2. They show generally very high correlations, with a lowest value of 0.5 (corresponding to p = 0.005, Fig. 2c). This implies that the interannual variability is very well reproduced in WRF. This is despite the fact that many of these stations are situated in urban environments, with a potential heat island effect, a lack of evaporative cooling that is seen for irrigated agriculture, and a very difference surface energy balance than snow-covered areas. Hence, their locations might not be representative of

the wider area, which might give rise to biases and trend differences when comparing the stations to the model outcome.

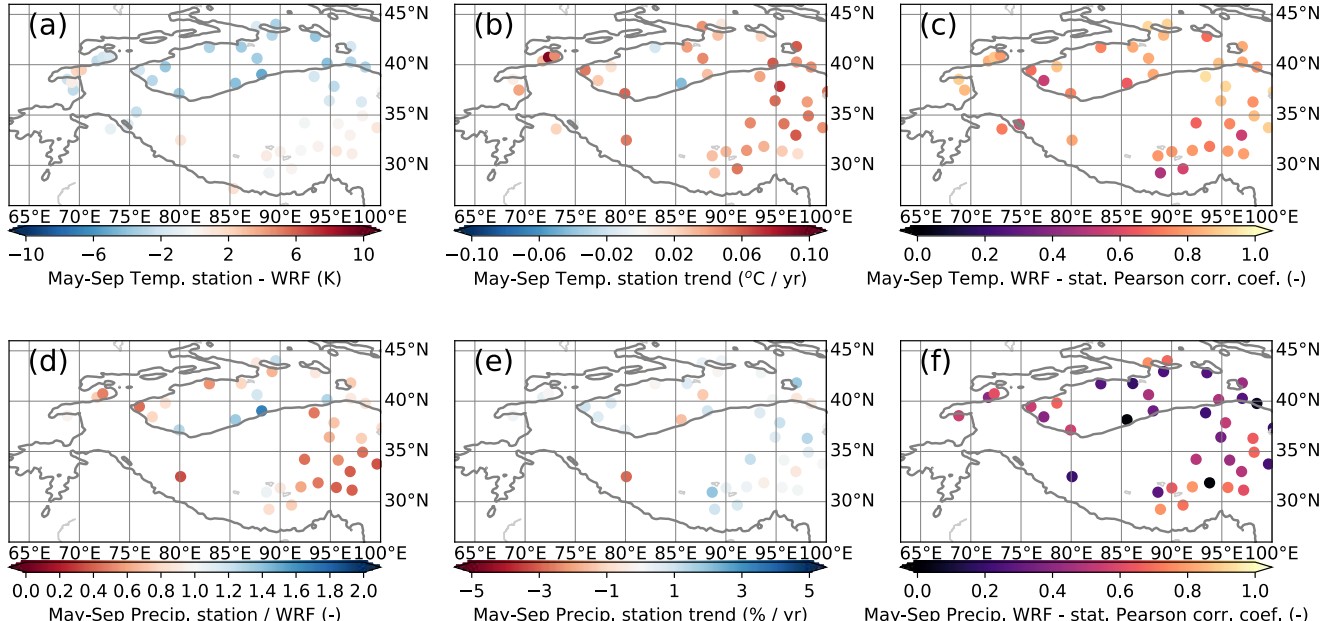

**Figure 2: Comparisons between 1980-2010 time series of station data and nearest WRF grid** point **for May-September temperatures (a-c) and May-September precipitation (d-f). Columns show temperature bias (a) and precipitation multiplication factor (d),** station trends **(b,e) and Pearson correlation coefficients. The 2000 m-contour is indicated by a solid line**

The stations in Fig. 2 closest to WKSK are almost exclusively in very arid regions, with a significant fraction of snowfall, which is more difficult to reliably measure than rain (Archer, 1998), making comparisons of precipitation very uncertain. Fig. 2 shows the comparison between time series of May-September precipitation, to limit the effect of snowfall. The stations show an increasing trend in May-September precipitation in the western Tarim basin and most of the eastern Tibetan Plateau (Fig. 2e). Our WRF output is generally wetter than what is measured at the stations, except some locations in the Tarim basin (Fig. 2d). The median root-mean-square deviation between WRF and the stations for the time-series of seasonal mean precipitation is 11.4 mm per month. The stations show that most of the Tarim basin and Tibetan Plateau are seeing an increase in May-September precipitation. The interannual variations are not represented by WRF as well as they are for temperature, but still show reasonable correlations for most stations, with values around 0.6 (Fig. 2f).

We also compare our WRF simulations with three similar data products with relatively high spatial resolutions, which have recently become available. We do note that all these datasets suffer from the lack of ground truth in WKSK, which means we cannot determine which dataset performs best in this region.

ERA5 is the follow-up of ERA-Interim (Copernicus Climate Change Service, 2017), with an improved spatial resolution of 0.25°, an improved temporal resolution, a more appropriate model input for e.g. sea surface temperatures, and more assimilated data. ERA5-Land is atmospherically forced by ERA5, and provides an even higher spatial resolution (0.1°) for land surface properties (Anon, 2019). Finally, we include the HAR dataset with a resolution of 10 x 10 km, which uses WRF to downscale

the NCEP FNR reanalysis dataset and re-initialises every day (Maussion et al., 2014). We compare temperatures between May-September, and annual precipitation, which give an indication of the parameters that are most relevant for glacier mass balance modelling. Because of the limited time overlap between the different datasets, we could only fully compare the period 2001-2010.

We binned all data to the same 0.5° x 0.5° grid to allow direct comparison. The mean values, trends, and interannual variability are compared in Figs. 3 and 4. It shows that ERA5 and ERA5-Land are nearly identical, and we only refer to ERA5 below. Our WRF model yields a warmer Karakoram than the other three datasets. Generally, the mean temperature differences are relatively minor, except for a warmer Tarim basin compared to HAR. We find very similar temperature trends as ERA5, although with smaller magnitudes. The magnitudes of the trends are also generally smaller than those in the station data (Fig.

2**b**). The WRF interannual temperature variations correlate very well with ERA5, except two areas in the Tarim and the inner Tibetan Plateau. This is not surprising, given that our WRF model is forced by the similar ERA-Interim data. The whole western part of HMA, including WKSK, is especially well-correlated to ERA5. In that region, the correlation with HAR is weaker, but the correlation between HAR and our WRF data is very strong in East HMA. The differences with HAR might be explained by the different forcing, or by the difference in used physics modules, but this requires further study.


Differences between datasets are larger for precipitation, at least for the mean values and interannual variability. Our WRF simulations give results that are relatively wet in the Karakoram, and relatively dry in the Himalaya. However, the precipitation trends are very similar to ERA5 in both pattern and magnitude. An exception is the arid Tarim basin, which has an increasing trend in WRF, but a decreasing trend in ERA5. HAR shows a positive precipitation trend in most of HMA, with a very high

trend in the Tarim basin. The correlation of the interannual variability is low in WKSK and parts of Tien Shan, which could be explained by the relatively high influence of the irrigated areas in the Tarim basin on the annual precipitation (Fig. 3 of de Kok et al., 2018). Since our WRF model outcome is the only one of the four datasets that explicitly includes irrigation, this could explain the difference in annual variability.

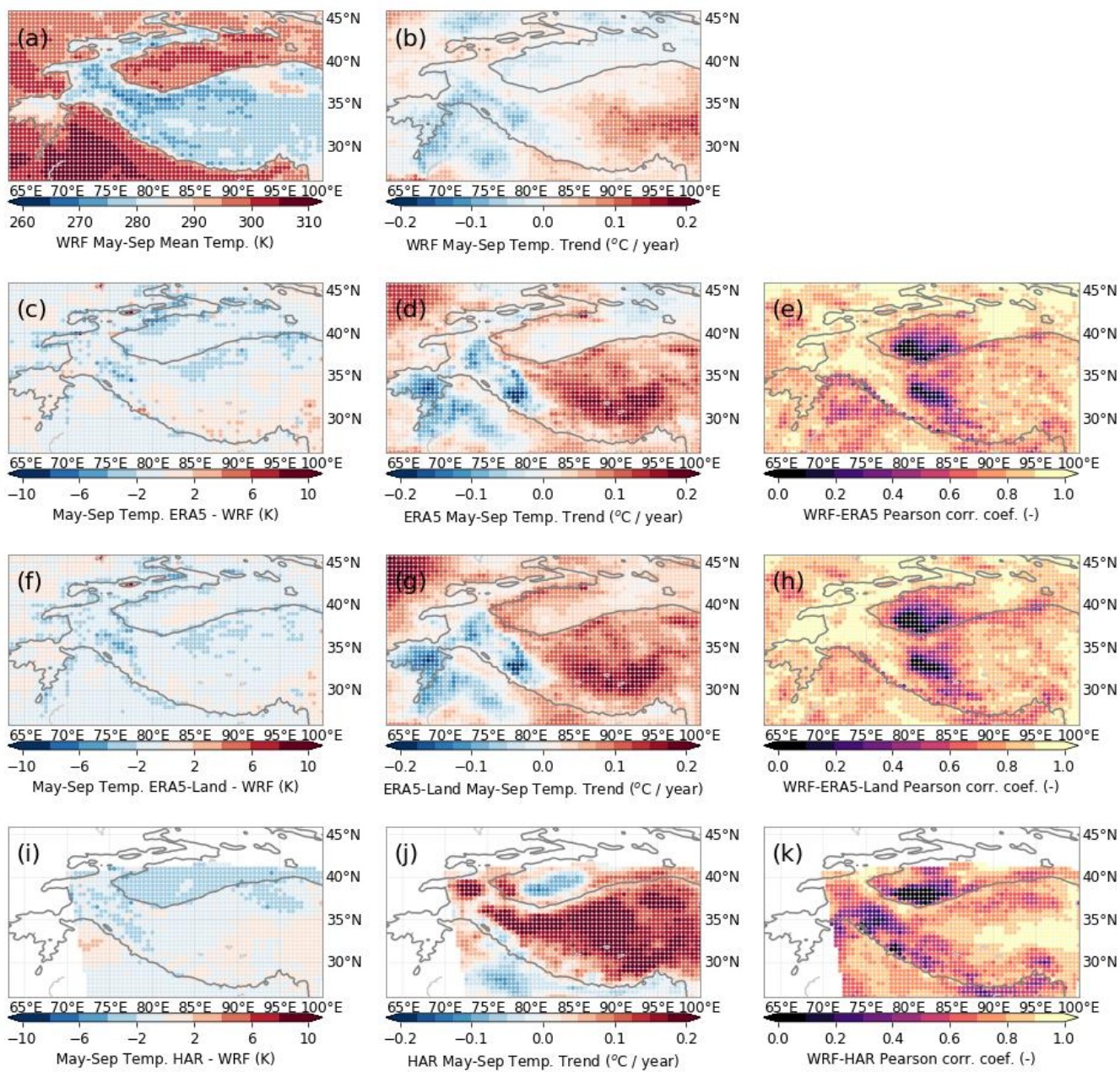


**Figure 3: Comparison of WRF temperature output [a-b] with three other datasets (ERA5 [c-e], ERA5-Land [f-h], and HAR [i-k]). Columns show biases (c,f,i) with respect to the May-September mean temperature (a), May-September temperature trends (b,d,g,j), and Pearson correlation coefficients between the datasets and our WRF results (e,h,k). The 2000 m elevation contour is indicated by a solid line.**


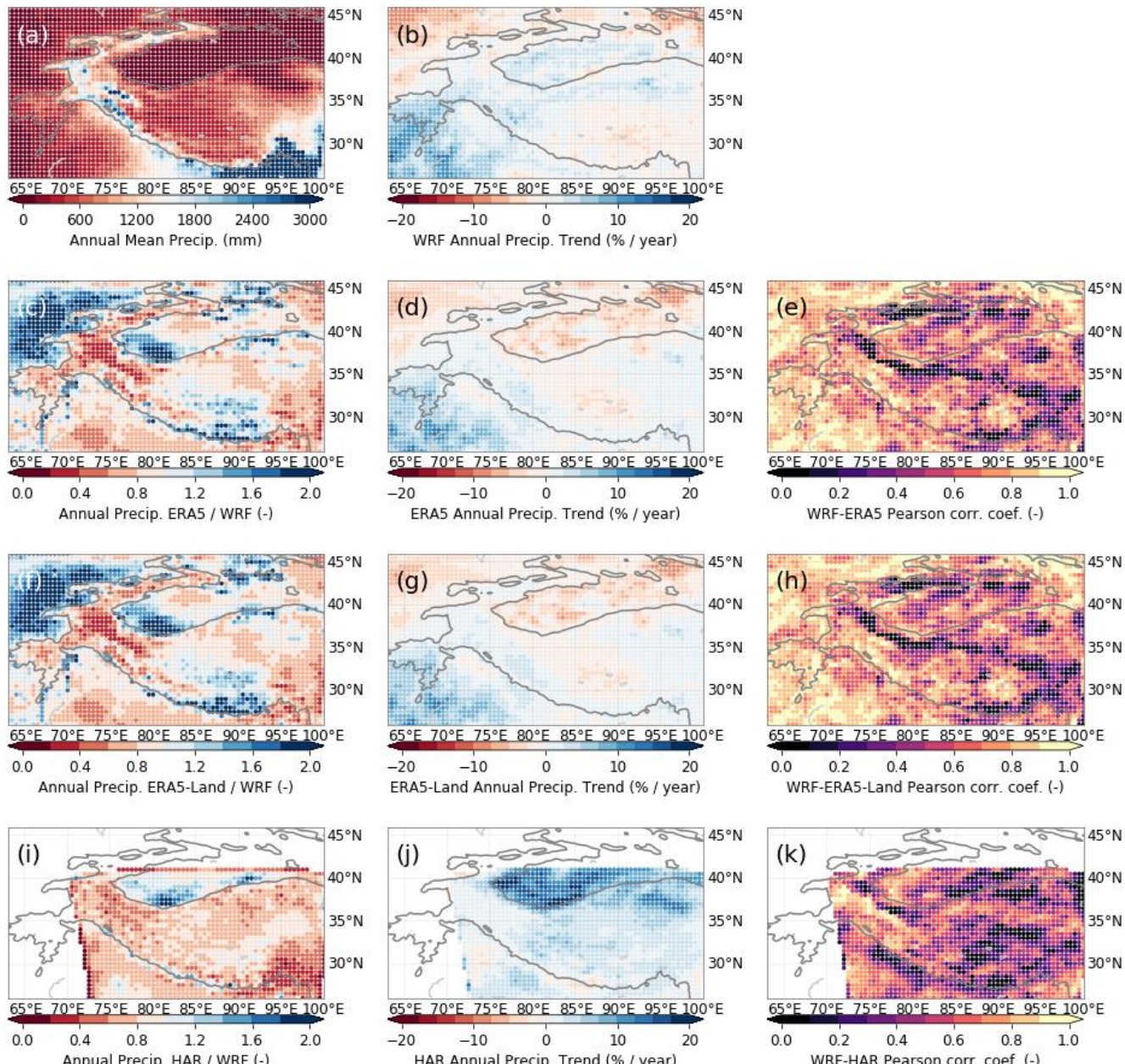

**Figure 4: Comparison of WRF precipitation output [a-b] with three other datasets (ERA5 [c-e], ERA5-Land [f-h], and HAR [i-k]). Columns show precipitation multiplication factors (c,f,i) with respect to the annual mean precipitation (a), annual precipitation trends (b,d,g,j), and Pearson correlation coefficients between the datasets and our WRF results (e,h,k). The 2000 m elevation contour is indicated by a solid line.**

The station and reanalysis data show a good agreement with our WRF output in many locations, but the comparison is hampered in WKSK due to the aforementioned fundamental uncertainties. Remote sensing data can also be used for comparison, but also there, uncertainties can be very high. This is especially true for precipitation measurements in mountainous areas, but also other remote surface measurements of relevant parameters are uncertain in mountainous areas (Lundquist et al., 2019). However, the atmosphere above the mountains can be measured with some confidence. Especially the atmospheric humidity can be used to increase the confidence in the interannual variability of the precipitation, since the two are strongly related. Here, we compare retrieved atmospheric humidity from AIRS and AMSU data (AIRS Science Team and Teixeira, 2013) above the mountains with humidity from our WRF output. These retrievals determine the humidity from satellite measurements at wavelengths in the infrared and microwave with very limited assumptions (Susskind et al., 2014), and hence can be considered as good validation dataset. Figure 5 shows the comparison of the mean AIRS specific humidity between May-September and between 400-500 hPa, and the corresponding WRF specific humidity interpolated at the middle of this layer (447.2 hPa) for the overlapping years 2003-2010, binned at the AIRS-AMSU resolution. The two datasets show a very high overall agreement, both in the patterns of humidity trends, as well as the correlation of interannual variability, with correlation coefficients generally above 0.9. This analysis shows that the moisture transport in our WRF model closely follows what we know of the atmosphere around WKSK. Near the edges of our modelling domain, our errors are naturally larger.

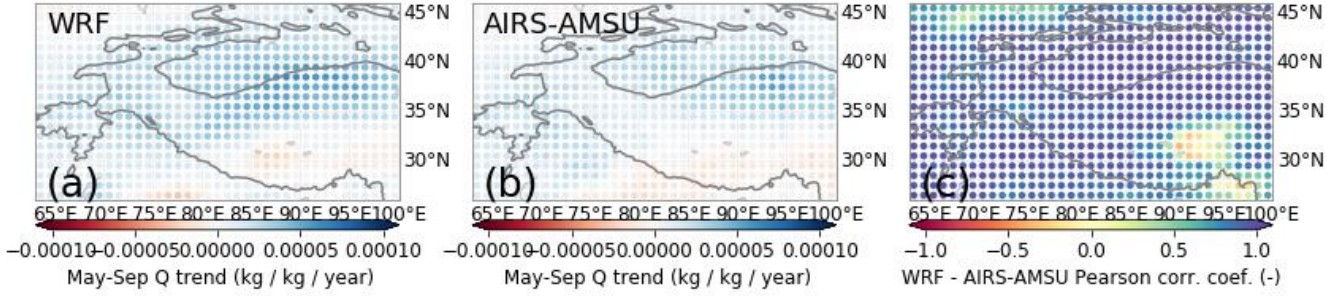

**Figure 5: May-September mean specific humidity trends at 447.2 hPa for WRF (a), and between 400-500 hPa for AIRS-AMSU (b) between 2003-2010. Panel (c) shows the Pearson correlation coefficient between them.**

Another variable that is important in our model is evapotranspiration. It cannot be directly measured remotely, but there are several datasets that calculate it from other remotely sensed products, either directly or through data assimilation. These datasets are all validated to some extent, but vary greatly nevertheless, as we illustrate in Fig. 6 for July 2010. We show evapotranspiration from GLEAM v 3.3a (Martens et al., 2017; Miralles et al., 2010), which assimilates various soil moisture, temperature, radiation, and precipitation products. Furthermore, we show SSEBop (Senay, 2018) data, which uses MODIS temperatures directly, ERA-Interim reanalysis data, and our WRF output. On the inner Tibetan Plateau, the WRF output agrees very well with the GLEAM data. Interannual variations also match very well between WRF and GLEAM in snow-free areas

on the Tibetan Plateau, with correlation coefficients above 0.5 for time series between 1980-2010. However, it is clear that GLEAM does not represent the irrigated areas well, with evapotranspiration in heavily irrigated arid regions in July that is as low as the surrounding deserts in e.g. Tarim and Indus basins, which is not realistic. In contrast, SSEBop shows very high evapotranspiration in the irrigated regions. The WRF output better resembles SSEBOP in those areas, although generally has lower maxima, which are only in part explained by the difference in spatial resolution, as is evident from e.g. averaging over 1x1° areas. ERA-Interim does not show the irrigation as prominently as WRF or SSEBop, but has a generally higher evapotranspiration values over unirrigated areas, such as the Tibetan Plateau. In general, the WRF simulated evapotranspiration is intermediate compared to the other datasets with plausible spatial patterns and magnitudes. However, the figure illustrates the problem with the high uncertainty in evapotranspiration over large areas in and around HMA.

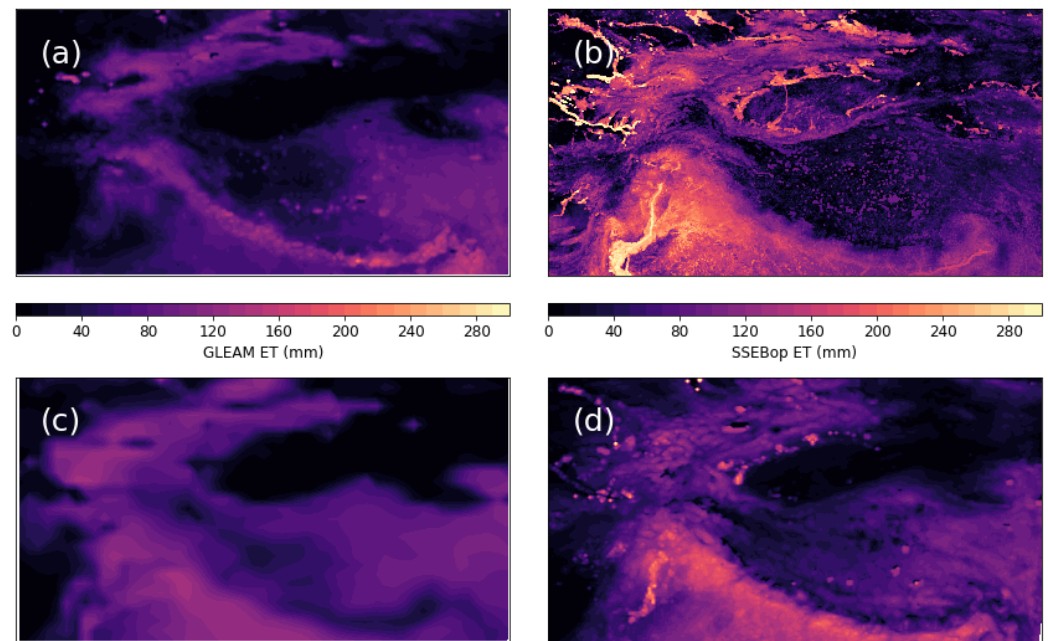

**Figure 6: Evapotranspiration for July 2010 from GLEAM (a), SSEBop (b), ERA-Interim (c), and WRF (d). Mean values for the plotted domains are: 44 mm (a), 46 mm (b), 57 mm (c), and 59 mm (d).**

We further investigate the realism of the effect of irrigation in our model by comparing remotely sensed surface specific humidity from AIRS and AMSU retrieval with our WRF specific humidity at 2 metres. These are not exactly the same quantities, as AIRS has a finite vertical resolution, but the variations over time can be compared. We focus on the irrigated area in the Tarim basin, close to the Kunlun Shan, which is the most important in the later discussion on the Karakoram anomaly. The flat terrain makes the retrievals near the surface more certain compared to mountainous regions, where altitude,

and hence pressure and humidity, strongly vary within the spatial resolution of the measurements. The comparison between means over May-September for 2003-2010 is shown in Fig. 7. Even though we did not nudge WRF towards ERA-Interim near the surface, the model still follows the humidity observations in the irrigated region in the Tarim basin very closely, with a Pearson correlation coefficient of 0.97. This gives further confidence that the irrigation we apply there is not unrealistic.

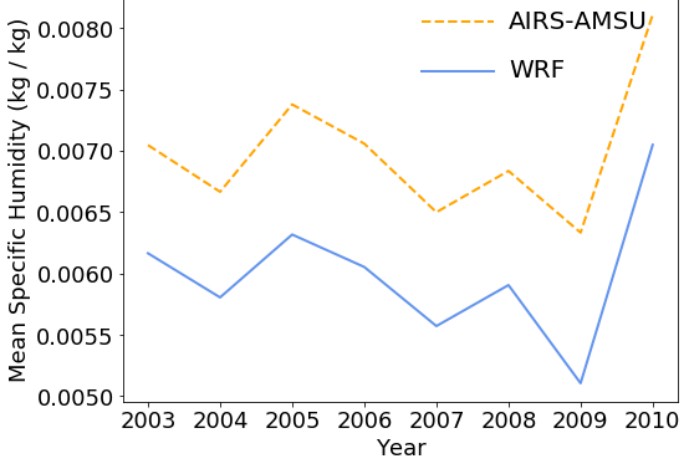


**Figure 7: May-September mean specific humidity at 2 metres from WRF (blue, solid line) and AIRS-AMSU surface humidity (orange, dashed line) for a 1° x 1° bin around 38.5° N, 77.5° E, which is an irrigated area in the Tarim that contributes to the snowfall in WKSK.**

### 3.2 Climatic trends

To get an impression how glaciers might have been affected by changes in the climate, we illustrate the trends for two relevant variables: the 2m-temperature in the melt season and the annual snowfall (Fig. 8, see also Fig. 9 for representative time series). For each grid point, the melt season was defined as the months where the mean daily temperature is above -5 °C, since for these months temperatures will likely be above freezing at least part of the time. A threshold value of 0 °C slightly increased the positive temperature trends at lower elevations in WKSK, but meant no trends for the highest elevations could be

determined. The trends show that temperatures in the melt season have generally increased, with the northern part of the domain heating up the fastest and parts of the Indo-Gangetic Plain, Kunlun Shan, Karakoram, and the Tibetan Plateau showing only modest increases in temperature. The temperature increase is there despite a recent decrease in summer temperatures in the region (Fig. 3). Fig. 9 shows that the trend and the interannual variability of temperature are very similar for nearby regions

of both growing and shrinking glaciers. The snowfall trends in Fig. 8 have a very different pattern, with most of the Tibetan
Plateau showing an increase and the western and southern mountain ranges, such as the Himalaya and the Hindu-Kush,
showing a decrease in snowfall. Furthermore, the mean level, the trend, and the interannual variability of snowfall is quite
distinct for the two nearby regions of contrasting glacier mass balance trends. The increase in snowfall in WKSK mainly occurs
in May, June, and September, whereas the decrease of snowfall in southwestern HMA occurs mainly in March (see Fig. 10**d**
for region averages).


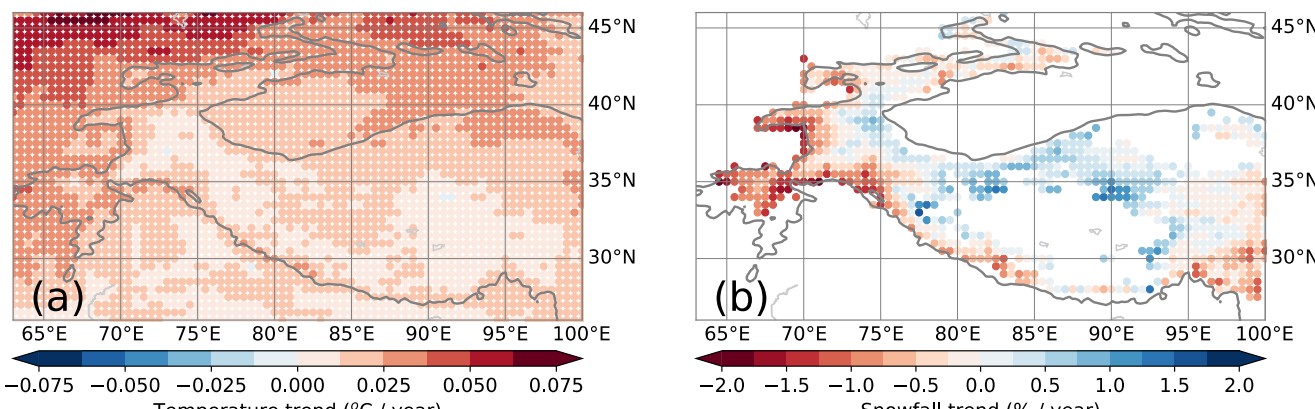

**Figure 8: Trends between 1980-2010 of temperature in the melt season (a) and annual snowfall (b), averaged over 0.5x0.5° bins for
clarity. Regions with monthly snowfall of less than 10 mm were masked out. The 2000 m elevation contour is indicated by a solid
line.**

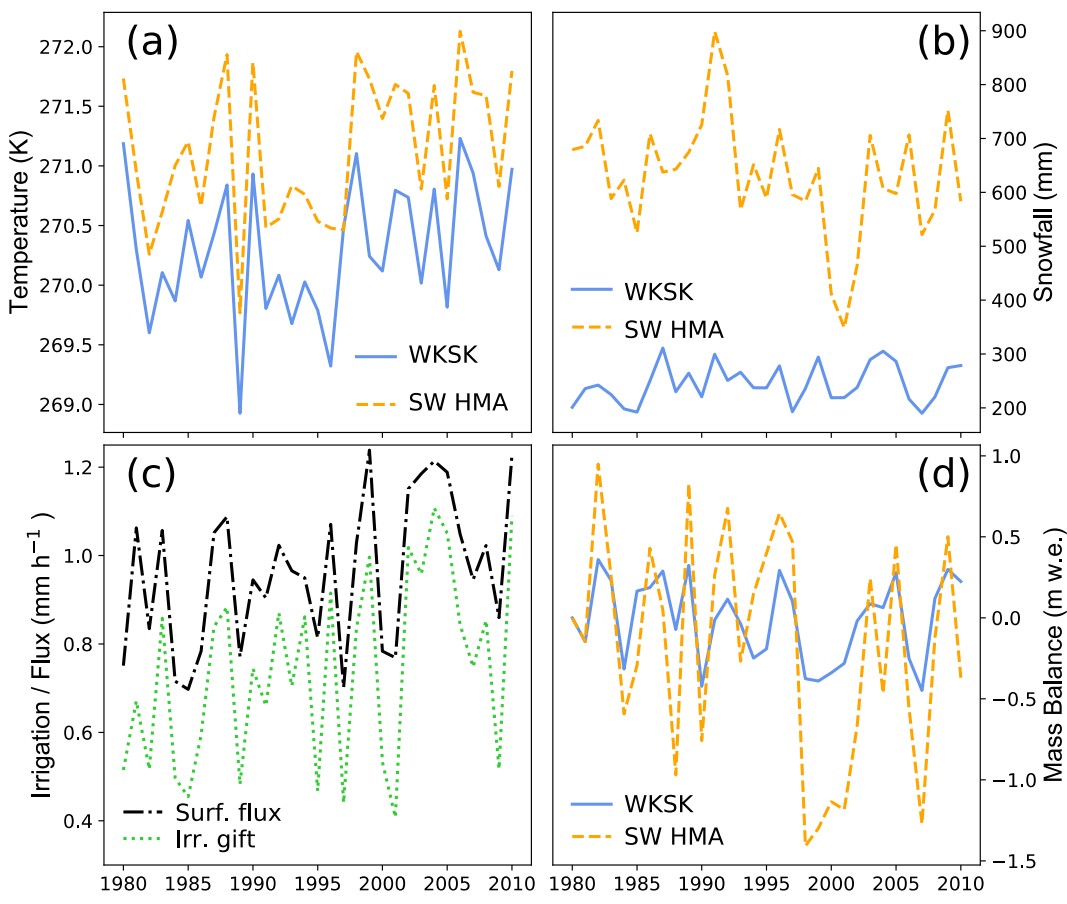


**Figure 9:** Time series of annual mean temperature (a), annual snowfall (b), mass balance (d) for two nearby 2x3° bins in WKSK and southwestern HMA that have, on average, growing glaciers (38-40° N, 73-76° E, blue lines) and shrinking glaciers (35-37° N, 72-75° E, orange, dashed lines). Panel *c* shows the time series of annual irrigation gift (green, dotted line) and annual surface moisture flux (black, dot-dashed line) for the most heavily irrigated point in the Tarim.

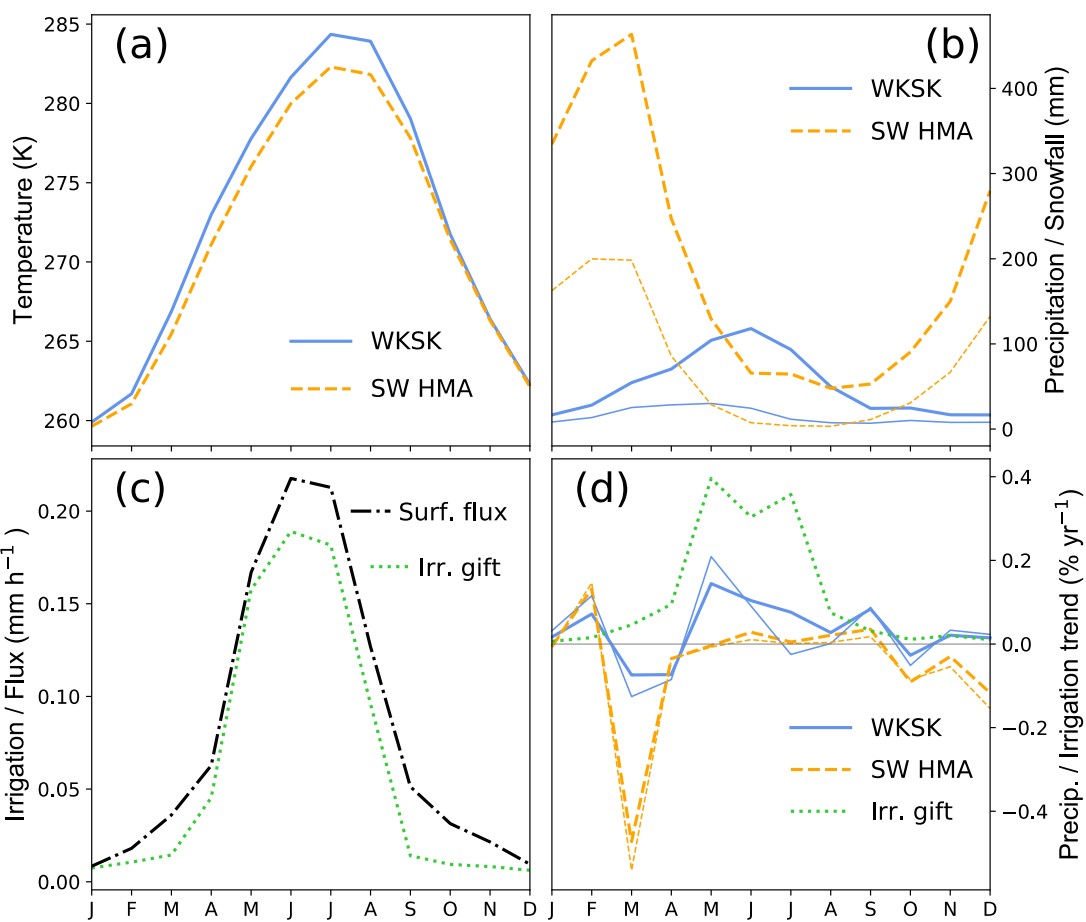

**Figure 10: a)** Mean seasonal cycle of temperature and **b)** precipitation (thick lines) and snowfall (thin lines) between 1980-2010 for the WKSK (blue lines) and southwestern HMA (orange, dashed lines), as shown in Fig. 14b,d. **c)** Mean seasonal cycle of the irrigation gift (green, dotted line) and surface moisture flux (black, dot-dashed line) of the most heavily irrigated point in the Tarim. **d)** Trends in precipitation (thick lines) and snowfall (thin lines) for the WKSK (blue lines) and southwestern HMA (orange, dashed lines), and the trend in irrigation from the most heavily irrigated point in the Tarim (green, dotted line), all as percentages of the annual mean value.

### 3.3 Glacier mass balances

The resulting pattern of simulated mass balance (Fig. 11) shows a strong resemblance to the measured pattern of mass balances of recent decades. Most notably, we also obtain growing glaciers in WKSK, whereas the glaciers in other regions show large mass losses. In fact, all points where we model glacier growth in Fig. 11a also show growth or stable conditions in observations (Brun et al., 2017; Kääb et al., 2015), except one point in Kääb et al., (2015). A more detailed quantitative comparison of the above results and the observed mass balances is hampered by the fact that our simulations only go out to 2010, and hence we cannot compare with the most recent, and most accurate geodetic mass balance data. However, we compare our results for the intermediate period 2000-2008, as presented by Brun et al. (2017), in Fig. 12. The results generally match reasonably well,

although our model seems to show too little growth for the growing glaciers. However, note that the errors on these observations (Brun et al., 2017) are large (~0.3 m w.e.). Furthermore, both the climate model and the glacier model will be associated with errors. However, in both cases the growing glaciers are only present in the same region, mainly WKSK and the Tibetan Plateau. By modulating the initial mass balance in the model, we find that on average 41% of the modelled mass balance in 2010 is determined by the initial mass balance in 1980. Although the mass balances in 1980 were observed to be

less extreme than in the 21st century (Bolch et al., 2012; Maurer et al., 2019), parts of HMA already had negative mass balances then, with the magnitude of initial mass balances generally less than 0.4 m w.e. yr$^{-1}$. This would result in an error on the mass balances in Fig. 10 of less than 0.2 m w.e. yr$^{-1}$. Despite these uncertainties, our results clearly show that the climatic change of the recent decades has favoured growth of the glaciers in the regions where actual growth is observed, and not in the places where glaciers are melting fastest.


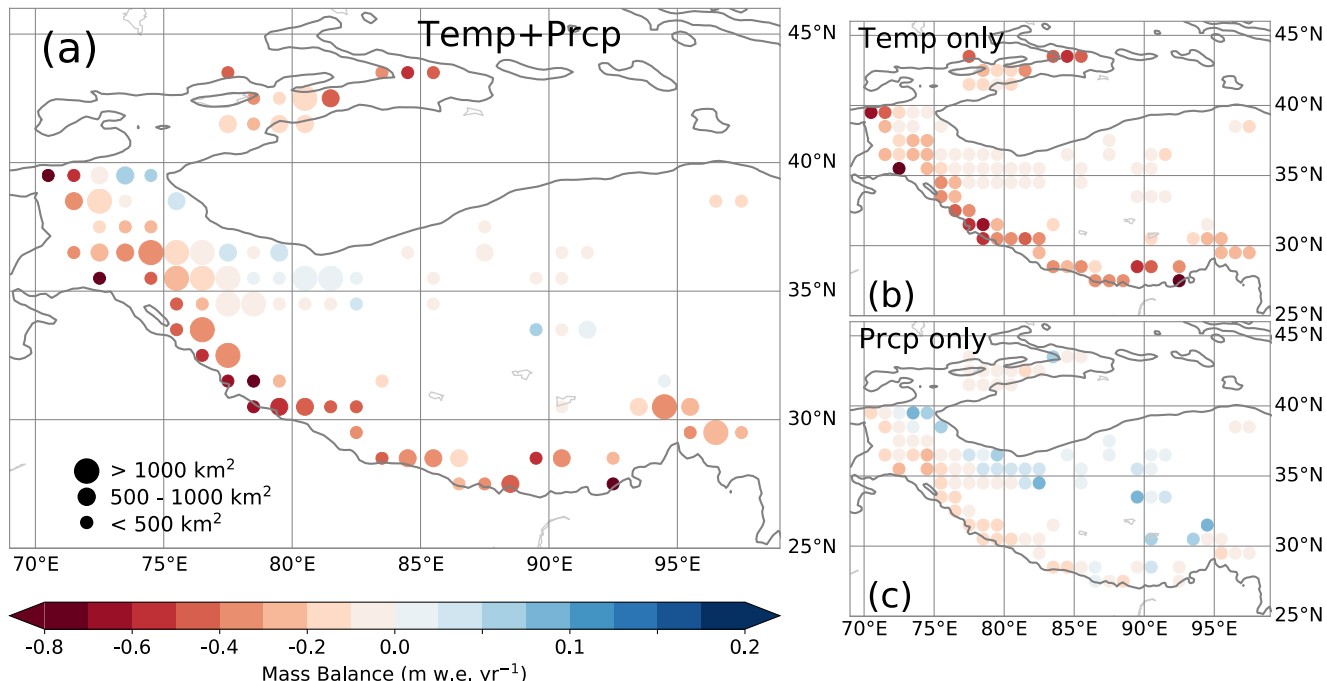

**Figure 11: Simulated mean mass balance between 2000-2010 forced by changes in temperature and annual snowfall from WRF (a), only changes in temperature from WRF (b), and only changes in annual snowfall from WRF (c). Results are binned in 1x1° bins, and bins with total glacier volumes less than 5 km³ are not shown, to enable comparison with previous studies. The 2000 m elevation**
**contour is indicated by a solid line.**

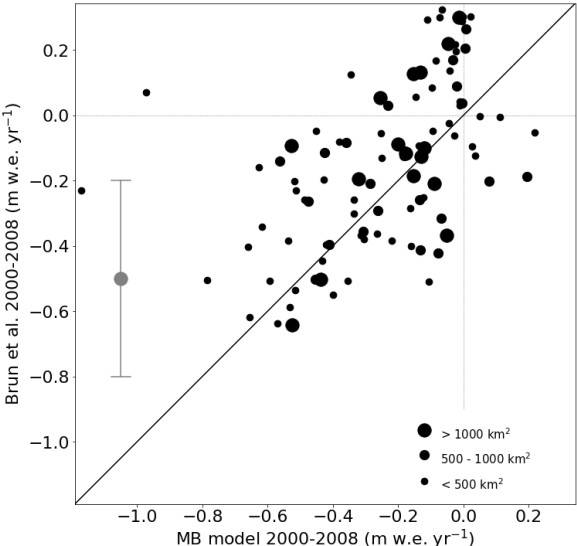

**Figure 12: Comparison between mean modelled mass balances from this work, binned on a 1x1° grid as in Fig. 11, and those derived from observations of Brun et al. (2017), which are on the same grid, between 2000-2008. The size of the mean errors on the observed mass balances is illustrated by the grey error bar.**


We also ran the glacier mass balance model forced by changes in temperature or snowfall only, to disentangle the model sensitivities of the two different variables on the glacier mass balances (Figures 11b and 11c). These results show that the glaciers in the western and southern HMA mainly lose mass due to the increase in temperature, while the decrease in precipitation gives a much smaller mass balance response in this region. On the other hand, in the regions where the glaciers

are growing, the glaciers are barely affected by the temperature increases in our model. The glacier growth in these regions is mainly caused by an increase in snowfall (Fig. 11c). Furthermore, the increase in snow is possibly also responsible for moderating the temperature increases due to the high albedo of fresh snow, which leads to less energy being used for melt. However, the weak temperature response in WKSK is not only caused by the limited temperature trends, but is also due to the limited glacier temperature sensitivity there. We demonstrate this by forcing the glacier model with uniform temperature and

precipitation trends (Fig. 13). The reduced temperature sensitivity is in line with previous work (Sakai and Fujita, 2017; Wang, et al., 2019), which argue that the generally large masses of the glaciers, and high equilibrium line altitudes, are important in explaining the lower temperature sensitivity in WKSK. The decrease in snowfall in the western and southern HMA has a far smaller impact on the mass balance than the increase in temperature. Especially the Himalaya show a low sensitivity to precipitation (Fig. 13). To be able to model thousands of glaciers, our mass balance model is relatively simple and does not

solve the full energy balance. A full energy balance model at 1 km resolution has shown that the temperature increases can amplify melt in the monsoon-dominated Himalaya, whereas snowfall increases in the melt season can amplify glacier growth in the Karakoram (Bonekamp et al., 2019). Hence, more detailed models will likely strengthen our conclusion that the observed mass gains are caused by snow increases, whereas the observed mass losses are mainly caused by temperature increases.

Unfortunately, modelling the climate and glaciers of the entire HMA at a sub-kilometre resolution for 30 years is currently
beyond our capabilities.

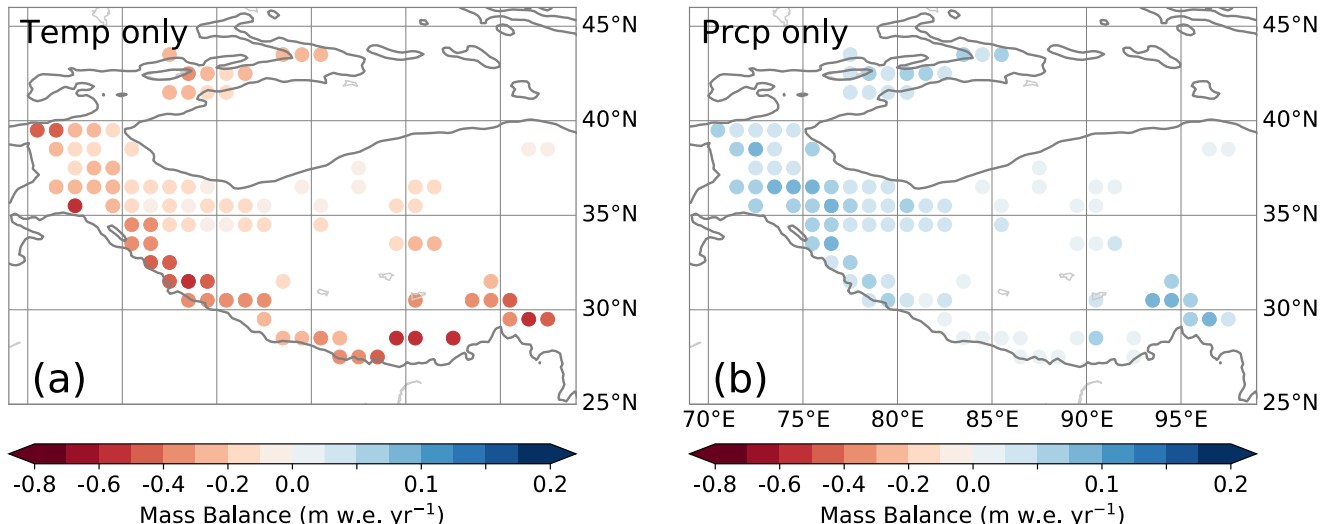

**Figure 13: Simulated mean mass balance between 2000-2010 forced by a spatially uniform and constant temperature increase of +0.01 °C yr⁻¹, with snowfall kept constant (a), and a spatially uniform and constant snowfall increase of +0.5% yr⁻¹ of the annual**
**mean value, with temperature kept constant (b). Panels *a* and *b* thus show the relative sensitivity to temperature and snowfall, respectively.**

### 3.4 Moisture sources

The trends in moisture source regions for WKSK (Fig. 14a,b) indicate that the largest increases in moisture from a given source to precipitation in WKSK occur in the mountains themselves. This increase in recycling occurs mainly in May, and is also the
main cause of the increase in precipitation in September (see Fig. 15). The increase in recycling is probably a natural consequence of the increased precipitation there. The regions with the second largest increases are the areas in the Tarim basin where irrigation has increased the most, which contributes mainly in May-July, with May showing the largest resulting increase in snowfall (see Figs. 10 and 15). In July, the increase in Tarim irrigation still contributes to increasing precipitation in WKSK, but it falls more in the form of rain, compared to May, where it is mainly snow (Fig. 10). Another region that contributes to
the increase in precipitation in WKSK is the Junggar basin, northeast of the Tarim basin. This is another arid region that has experienced rapid increases in irrigation. The increases per grid point are lower there, but they are spread out over a larger area. A final source region with an overall large positive trend is the Caspian Sea and the Caucasus. Note again that, due to a systematic offset in surface moisture flux between WRF and ERA-Interim, the moisture source trends in the Tarim and HMA are underestimated with respect to the other regions.


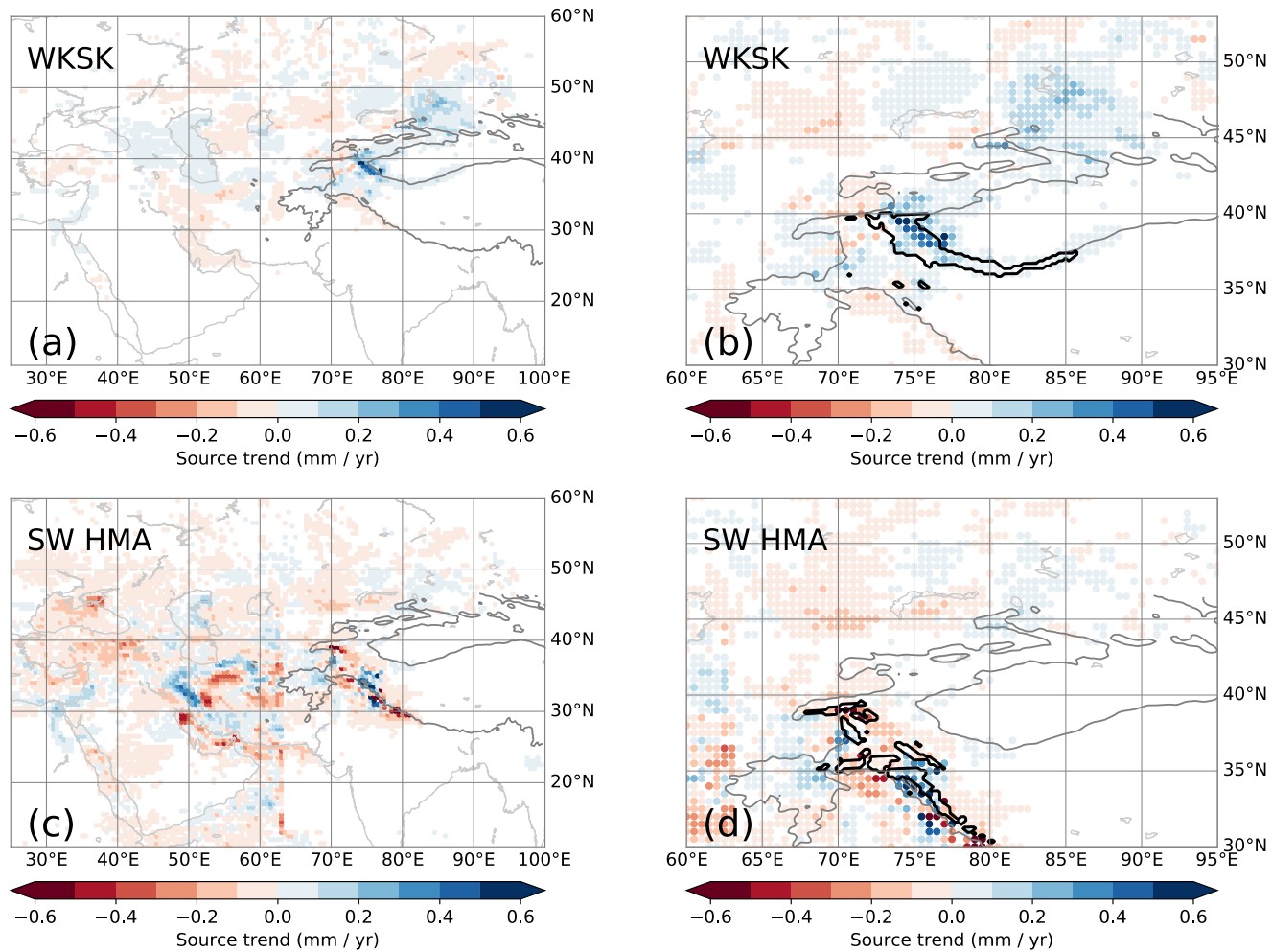

**Figure 14: Trends in the amount of moisture from a given source contributing to precipitation trends in the target area (a,c), with a detailed view (b,d) around the target area from which the parcels were released (contoured in bold) for WKSK (a,b) and southwestern HMA (c,d). Trends with absolute magnitudes smaller than 0.02 mm yr⁻¹ are made white. The 2000 m elevation contour in the WRF domain is indicated by a solid line.**

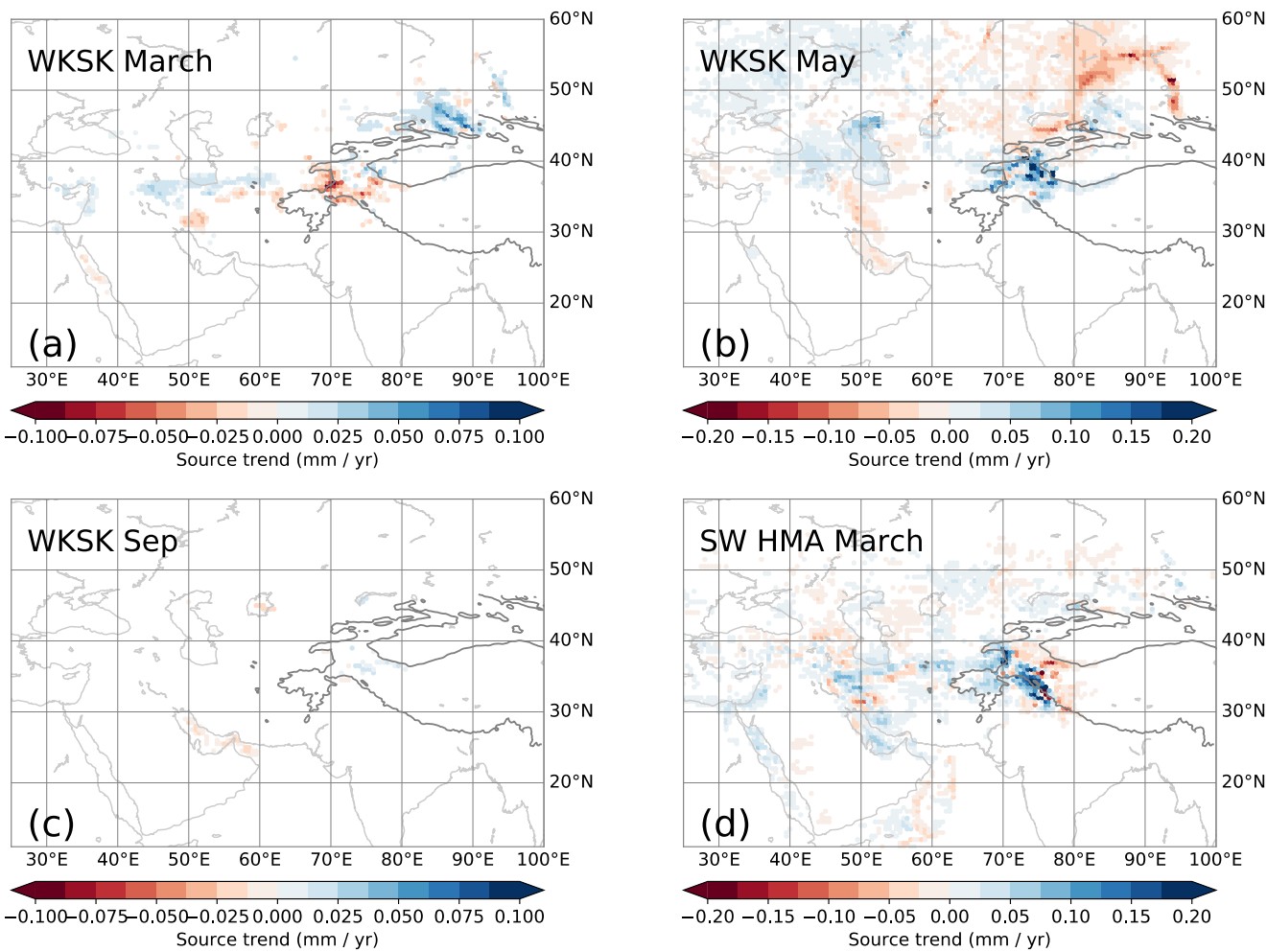

**Figure 15: Trends in the amount of moisture from a given source for WKSK for March (a), May (b), September (c) and for southwestern HMA in March (d). These months correspond to large negative or positive trends in snowfall in Fig. 10. Note the different scales.**


These results imply that evapotranspiration from irrigated areas in arid Northwest China play a large role in adding water to parts of HMA and hence to the observed positive mass balances. This is in line with recent work that shows that the recent wetting of Central Asia and the Tarim basin is associated with an increase in evapotranspiration in these regions (Dong et al., 2018; Peng et al., 2018; Peng and Zhou, 2017). The increase in the total evapotranspiration is influenced by the increase in

potential evapotranspiration (Fang et al., 2018), increase in water availability (Jian et al., 2018), and increase in irrigated land area. On the interannual timescale, precipitation in WKSK strongly correlates with the moisture source amount in the western Tarim basin (Pearson r=0.96 below 3500 m, r=0.68 for the entire WKSK, as indicated in Fig. 14b). A similar correlation exists between the WKSK precipitation and the Caspian Sea moisture source amount (r=0.89 below 3500 m, r=0.43 for the entire

WKSK), showing the importance of the large-scale weather patterns. For the Junggar basin, this correlation is weaker (r=0.65 below 3500 m, r=-0.14 for the entire WKSK), since this region contributes relatively more in winter (Fig. 15), when less snowfall reaches WKSK (Fig. 10).

When performing the moisture tracking for the southwestern part of HMA, where snowfall has generally decreased (Fig. 14c,d), also the Caspian Sea and the Junggar basin positively contribute to the snowfall trend, whereas for these ranges the Tarim basin does not contribute to the snowfall trend, with maximal trends in moisture sources of less than 0.1 mm yr$^{-1}$. These results show that the irrigated areas in the Tarim basin are especially important in influencing the moisture supply to the Western Kunlun Shan (de Kok et al., 2018).

## 4. Conclusion and discussion

Our simulations, based on ERA-Interim and GLDAS reanalysis data, indicate that an increase of snowfall and a low temperature sensitivity are the main reasons why glaciers are growing or stable in western Kunlun Shan and Karakoram. This is the first time that the observed pattern of glacier mass balances in HMA is reproduced in a consistent way. We show that such a pattern can be reproduced using relative changes in temperature and precipitation in recent decades. Since we used relative changes to force our glacier model, we are less influenced by errors in the absolute precipitation amounts, caused by our low resolution or by our choice of model physics. We illustrate this using WRF runs performed for de Kok et al. (2018) for May-September of two years. We ran WRF at two resolutions: at 20 km with same the physics settings as in this study, but without any nudging, and at 4 km, which is of high enough resolution to explicitly resolve convection and avoid the cumulus parameterisation. There are large local differences in precipitation between the two runs, mainly due to the difference in resolution. However, when the relative ratio of the precipitation is plotted for two years (Fig. 16), similar to what is used in the glacier model, the two set-ups give much more similar patterns. Snowfall gives very similar results, but we decided to show total precipitation, where total numbers and cumulus errors are expected to be even higher. The relative changes in precipitation do not markedly show the topography, in contrast to the individual precipitation fields. Rather, relatively large regions show similar interannual changes in the precipitation. The patterns of precipitation change also agree well between the 20 km results and the 4 km results, despite the very different treatment of the convection and the difference in topographic resolution. The differences between the scaling factor in the two cases can be of the order of tens of percent, which is much smaller than the difference in absolute precipitation amounts that would be needed to model the mass balance directly from the WRF fields. Also temperatures are mutually correlated over larger areas in WKSK (e.g. Forsythe et al., 2017) and the glacier mass balances in HMA also vary mainly over a large scale, suggesting that large-scale weather patterns are on average more important in controlling the interannual variability of temperature and precipitation than the differences between valleys. The use of relative changes in temperature and precipitation has thus made our results more robust against possible errors in the detailed treatment of the complex mountain meteorology.

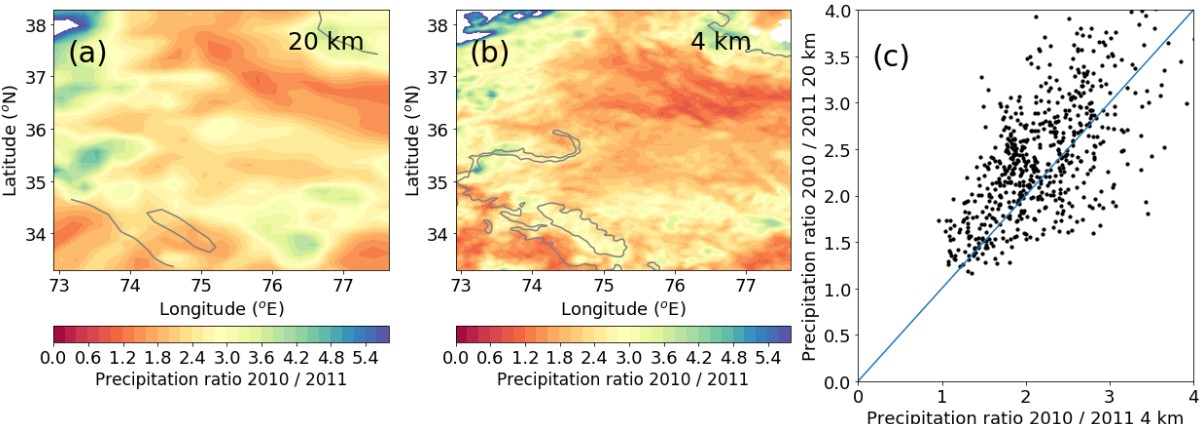

**Figure 16: Precipitation ratios between May-September of two years for the WRF run at 20 km, with cumulus parameterisation (a), that at 4 km resolution, without parameterisation (b), and the two compared, when binned at the resolution of the 20 km run (c). The 2000 m elevation contour is indicated by a solid line.**

One of our main sources of error is setting up the initial mass balance gradient, and our assumption that the glaciers are initially in balance. Due to the inertia of the glaciers, the initial condition has relatively large influence on the eventual mass balance decades later, as discussed above. Furthermore, any errors in the mass balance gradient, e.g. due to errors in the downscaling of ERA-Interim data, will affect the temperature and precipitation sensitivities presented here, but will have less impact on the overall pattern of mass balances in HMA, since they are mostly determined by the changes of temperature and precipitation.

Our snowfall trends between 1980-2010 show some similarities, but also major differences with respect to a similar WRF study that did not include irrigation and used another re-analysis dataset (Norris et al., 2018). For instance, our temperature trends do not exhibit the strong summer cooling at low altitudes (e.g. the Tarim basin), and are more in line with station data (Waqas and Athar, 2018; Xu et al., 2010) in that respect. However, contrasting precipitation trends in WKSK and southwestern HMA, similar to Fig. 8, are also present in ERA5 data and the Norris et al. study (see Farinotti et al., 2020). Although the interannual variability of temperature and precipitation is reasonably reproduced, and our precipitation trends are similar to those in other datasets, our model results are associated with uncertainties, which are partly irreconcilable due to a lack of *in situ* measurements in WKSK. Furthermore, different parameterisations in the regional climate model, different irrigation schemes, and different glacier models will likely yield slightly different results. Using an ensemble of such approaches could be used to assess the robustness of the results presented here in the future. Furthermore, detailed studies at smaller scales will give more insight into individual glacier behaviour. It is then also possible to use more complex glacier models, e.g. those that take into account the full energy balance.

The pattern of snowfall trends in Fig. 8b roughly matches the precipitation pattern that is expected from an increasing influence of summer westerlies, as shown by Mölg et al. (2017). From this similarity, one could wonder whether the snowfall pattern from Fig. 8b is mainly caused by summer westerlies. These summer westerlies are also associated with strong heating and drying trends of the Indus Basin. An increase in irrigation also produces a very similar precipitation pattern as the pattern for summer westerlies, yet causes a cooling and wetting of the Indus Basin (de Kok et al., 2018). Our JAS trends of near-surface temperature and specific humidity from WRF (Fig. 17) indicate mostly cooling and wetting trends in the Indus basin, which is more in line with the increase in irrigation than with the increase in summer westerlies. ERA5 data for JJA also indicates a similarly strong irrigation effect in the Indus basin (Farinotti et al., 2020), as indicated by a wetting and cooling trend. The moisture tracking results (Figs. 14 and 15) indicate that much of the additional snowfall occurs in spring and summer, and originates from the East, with a large role for the irrigated areas. The decrease in precipitation in southwestern HMA is also clearly associated with westerly winds in winter, but not those in summer (see Figs. 10d and 14c). The pattern of snowfall trends in Fig. 8b is thus not only the result of changes in summer. When only JAS is considered, the pattern of precipitation trends look different from the annual snowfall trends (Fig. 17b). Therefore, the summer westerlies are likely not the main driver for the snowfall pattern seen in Fig. 8b. However, the May westerlies clearly have an important role in transporting the increase in evaporation from the Caspian Sea (Chen et al., 2017) to WKSK. Besides the Caspian Sea, the westerlies are mainly associated with a decrease in snowfall when the whole year is considered (Fig. 14a).

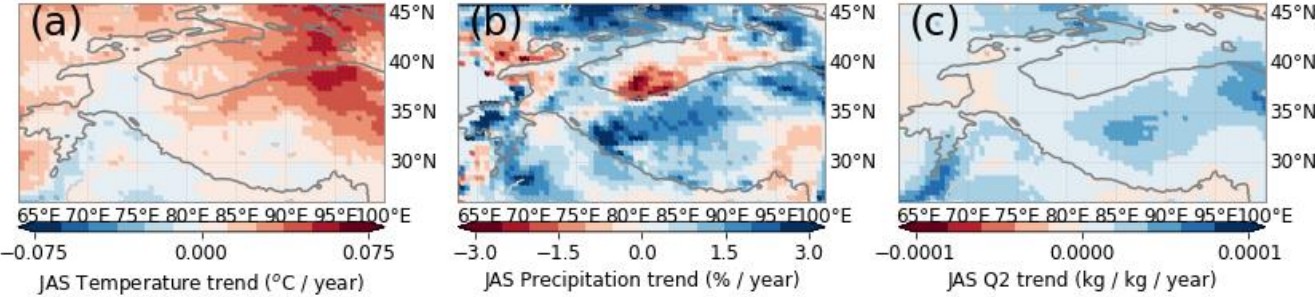

**Figure 17: WRF trends between 1980-2010 of near-surface temperature (a), total precipitation (b), and specific humidity (b) between July-September, averaged over 0.5x0.5° bins for clarity. The 2000 m elevation contour is indicated by a solid line.**

We show that the growing irrigated area in the arid region of Northwest China plays an important role in the increase in snowfall in WKSK. Previous studies have already shown that increases of irrigation in Northwest China can add precipitation to neighbouring mountains (Cai et al., 2019; de Kok et al., 2018), but we now show this process of increasing irrigation is also important compared to other changes in the atmosphere over the last few decades. Already before 1980, irrigation has increased in Northwest China (Fang et al., 2018), possibly contributing to the stable glacier conditions then. Future evolution of snowfall

in this part of HMA is partly linked to how the irrigated areas develop in the future. Changes in temperature, irrigated area, or irrigation efficiency are therefore important parameters in understanding future run-off from glaciers and snow in WKSK. The increase in water availability for irrigation in Northwest China might be partly the result of the loss of glacier mass in Tien Shan (Dong et al., 2018). The mass loss will first result in an increase in glacier melt run-off into the Tarim basin, but ultimately the run-off will decrease as the glaciers shrink to a small size (Kraaijenbrink et al., 2017). On the other hand, if the primary source of irrigation water is groundwater, the amount of irrigation for the region will also have a limited sustainable or economic level. Once the groundwater is depleted, our results suggest that the glaciers in WKSK will also receive less snowfall from this region, resulting in their retreat. The relative importance of groundwater extraction, melt from Tien Shan, and recycling from WKSK, for water availability in the Tarim is yet unknown and will require future study. Furthermore, improving the estimates of irrigation gifts, e.g. by remote sensing, could also improve the past climate reconstruction of WKSK. Greening and warming in West-Asia could provide additional snowfall to WKSK, together with an increase in westerly disturbances (Cannon et al., 2015; Kapnick et al., 2014), but if temperatures in HMA keep increasing, the increase in melt will probably counteract glacier growth in most of HMA in the long term. Our modelled mass balances show a decreasing mass balance trend for WKSK (Fig. 9d), but the trend is far too insignificant to draw conclusions about future mass balances. It is clear that the coupling between glacier mass balance, runoff, and irrigation in different regions creates a complex problem of water availability, which will need to be researched further to inform decision makers on irrigation policies.

**Acknowledgements**

We acknowledge funding from the European Research Council (ERC) under the European Union's Horizon 2020 research and innovation program (grant number 676819), the Netherlands Organisation for Scientific Research Innovational Research Incentives Schemes VIDI and VENI (016.181.308 and 016.171.019), and the Strategic Priority Research Program of Chinese Academy of Sciences (grant number. XDA20100300). Computing time was provided by the SURFsara CARTESIUS National Supercomputer of the Netherlands Organization for Scientific Research. We thank Rens van Beek for distribution of the PCR-GLOBWB irrigation data. We thank Fanny Brun and Jesse Norris for discussion. We thank Thomas Mölg, Dieter Scherer, and two anonymous reviewers for their careful reading and useful comments, which improved the manuscript.

**Code and data availability**

The data underlying our results in Figs. 8-15,17, i.e. monthly mean output from WRF of temperature and precipitation, annual glacier mass balances, and annual moisture sources, are directly accessible at *dataverse.nl* (https://hdl.handle.net/10411/ATONZD). Other data is available from the authors upon request. WRF and the glacier mass balance model are freely available. The moisture tracking model is available upon request from Obbe A. Tuinenburg.

## Author contributions

R.J.d.K. and W.W.I. designed the study, with input from all authors. R.J.d.K. performed the WRF modelling, P.D.A.K. performed the glacier mass balance modelling, and O.A.T. performed the moisture tracking. All authors contributed to the writing and editing of the manuscript.

## Competing interests

The authors declare that they have no conflict of interest.

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
