# Peer review of "Towards understanding the pattern of glacier mass balances in High Mountain Asia using regional climatic modelling"

_The Cryosphere, 2019_

## Author Comment (AC1) · 6 Nov 2019

We noticed that we accidentally placed the wrong figure as Figure 8 in the manuscript (panels b and c now show the same as Fig. 10). We apologise for this oversight and the inconvenience caused by it, and attach the correct figure here.

[Figure]

[Figure]

**Fig. 1.**

---

## Referee Comment (RC1) · Dieter Scherer (Referee) · 22 Nov 2019

Dieter Scherer (Referee)

dieter.scherer@tu-berlin.de

General comments

The authors of this study address the reasons behind the spatial patterns of glacier mass balance as observed in High Mountain Asia (HMA) over the last decades. While many of the glaciers in HMA follow the world-wide trend of losing mass, there are some regional exceptions like the so-called Karakoram anomaly, the Kunlun Shan and parts of Pamir, where glaciers are stable or even growing.

The authors used the Weather Research and Forecasting model (WRF) to downscale ERA-Interim reanalysis data from 1980-2010 to a domain of 20 km grid spacing covering HMA and adjacent regions, and then applied a glacier mass balance model forced

by the downscaling results. By this approach they have been able to reproduce the observed patterns of glacier mass balance in HMA. They found that the low temperature sensitivities of glaciers and increases in snowfall have caused positive mass balances in the western Kunlun Shan and Karakoram ranges (WKSK). They also detected that increases in snowfall are to large degree stemming from increases in evapotranspiration from irrigated agriculture in regions adjacent to HMA.

Dynamical downscaling was performed by using grid nudging of the upper 35 (of 50) vertical levels for horizontal wind, temperature, and humidity, which preserves the large-scale meteorological tendencies of the ERA-Interim reanalysis. The WRF model was re-initialised each month (plus 10 days of spin-up) with appropriate data reflecting seasonal changes in snow cover, surface and soil temperature, and surface moisture. Glacier mass balance was modelled by employing a glacier mass balance gradient model to all individual glaciers in HMA larger than 0.4 km$^2$ transiently for the period 1980-2010 assuming steady-state conditions in the beginning. They performed three separate glacier simulations using different forcings (precipitation and temperature, only temperature, and only snow) from the WRF data set. A moisture tracking algorithm was applied to identify the source areas of precipitation.

The study is comprehensive, well-performed and mostly well-presented in the manuscript, and the results are highly interesting. A mistake in one of the figures has already been corrected by the authors prior to my review. The study definitely deserves publication in The Cryosphere.

Specific comments

I do have only a few specific comments, which could be easily addressed by the authors.

First of all, the entire study depends on the accuracy of downscaled precipitation. It would therefore be of utmost interest to better understand the uncertainties in the WRF output. As the authors correctly state, in-situ meteorological observations are scarce,

and there is almost complete lack of data in the WKSK ranges, which makes it difficult to compare the WRF output with independent observations. This is especially true for high altitudes, i.e., the glacierized areas, where observational data are not available. Nevertheless, there are gridded data sets that could be used for comparison. Although they do not cover the entire study period (so far) and thus cannot substitute the ERA-Interim data used for downscaling, they could anyway be compared with the WRF results for shorter periods (as the authors have done with GLEAM data). The new ERA5 reanalysis and the High Asia Refined analysis (HAR) data set (Maussion et al., 2014) are suitable data sets in this respect. ERA5 data, especially the newest ERA5 land data set (https://cds.climate.copernicus.eu/cdsapp#!/dataset/reanalysis-era5-land?tab=overview), and the HAR data set do have very high spatial and temporal resolutions, such that they resolve mesoscale atmospheric processes, and thus orographically induced precipitation. HAR data are freely available at www.klima.tu-berlin.de/HAR. I would ask to authors to include a comparison of WRF output with these gridded data sets in the article. This could be put into a supplement with only a short paragraph in the main text.

The authors shall not only provide Pearson correlation coefficients but also further metrics like mean biases, r.m.s. deviations, regression slopes, etc., when comparing their WRF results with those from GHCN stations. I am not convinced that it is necessary to exclude so many GHCN stations by requesting at least 20 year of data coverage. This could be relaxed, or further comparisons may be added. I am also not convinced that it is sufficient to present results only for aggregated time periods, i.e., for annual mean air temperatures, May-September air temperatures, and July precipitation. Depending on the details of forcing the glacier model by WRF output, more detailed analyses of the WRF uncertainties are required, since snow- and ice melt can be rather variable from year to year, although years might have shown similar mean seasonal values for air temperature and precipitation.

In this respect, I would ask the authors to add more details on the WRF output and

its application for forcing the glacier model simulations and the moisture tracking algorithm. In particular, I would like to know the output time step (one hour?).

Reference

Maussion F, Scherer D, Mölg T, Collier E, Curio J, Finkelnburg R, 2014: Precipitation Seasonality and Variability over the Tibetan Plateau as Resolved by the High Asia Reanalysis. Journal of Climate 27(5), 1910-1927. https://journals.ametsoc.org/doi/full/10.1175/JCLI-D-13-00282.1

---

## Referee Comment (RC2) · Anonymous Referee #2 · 3 Dec 2019

General comments

In this study, a 20-km WRF downscaling of ERA-Interim is performed over High Mountain Asia over 1980-2010. The resulting data is much more representative of the temperature and snowfall trends over the mountains over this period than the reanalyses. Unlike previous studies, the WRF simulations include moisture in the land surface to reflect increases in irrigation over the simulation period, in particular over the Tarim Basin, in western China. The WRF output is used to force a glacier mass balance model over the given period and thus addresss the meteorological impacts on glacier mass balance in recent decades, in particular to address the Karakoram anomaly. It is shown that some glaciers have advanced in the western Kunlun Shan and Karakoram (WKSK) due to increased snowfall. By tracking the moisture that leads to precipitation

over the WKSK, the authors identify that increased evapotranspiration over the Tarim Basin and to a lesser extent over other areas has contributed to this increased snowfall. Therefore, the increased irrigation may have been a factor in the Karakoram anomaly.

This study is highly novel in: 1) using high-resolution model data, which incorporates the meteorological effects of increased irrigation, to force a glacier model; 2) reproducing the observed patterns of glacier mass balance. As such it should be published and will be of great interest to both meteorologists and glaciologists. I just have a few comments that I would suggest the authors consider before the manuscript is suitable for publication.

Specific comments

1) In various figures (2,4,8,10,11,12), concise panel labels would be very helpful to allow the reader to immediately see what each panel shows, without having to read the caption.

2) Similarly, in Figs. 6 and 7, a legend would be very helpful so the reader can immediately see what each line represents.

3) L54-55: By "the amount of irrigation needed to compensate evapotranspiration", do you mean, after subtracting actual precipitation?

4) L55: PCR-GLOBWB should be defined/spelled out.

5) L71-73: How are these concentrations of the various greenhouse gases fed into the model? Is it through the radiation scheme?

6) L76: What is meant by "convergence between months"?

7) L92: What is meant by "both deltas"?

8) L152-154: In this sentence, it sounds like the implicit assumption is that the measurements are biased, but assuming these biases are constant in time, then we can use them to evaluate WRF's interannual variability. This should be made explicit.

9) L156: GHCN has not been defined.

10) L159-160: Please explain the relevance of many of these stations being situated in urban environments.

11) L166-167: Implicit in this sentence is that the stations measure snow less reliably than rain. Please make this explicit and provide a reference.

12) L185: How do you know that the discrepancy is only in part due to the different spatial resolution? Have you quantified the effect of the spatial resolution?

13) Figure 6 is never referenced in a meaningful way. This figure shows nicely that there is no clear distinction between growing and shrinking glaciers in terms of temperature trends, but that there is a clear distinction in terms of snowfall trends. It would be nice to have some words to this effect in the text.

14) Figure 6 caption: how is representativeness of the bins determined?

15) L230-232: It is quite confusing when you say "our simulations only go out to 2010, but we compare our results for 2000-2008". Why not compare results up to 2010? If 2008 is as far as the observations go, then the limitation is in the observations, not the simulations.

16) L232-233: As well as the model showing too little growth for the growing glaciers, it shows different glaciers growing to the ones in the observations (Fig. 9). Are the growing glaciers in the model and observations at least in the same areas?

17) L259: Presumably the low glacier temperature sensitivity in the WKSK is because, even with warming, temperatures in in the WKSK are still generally below freezing? This could be clarified. Or if there is a different reason?

18) I am slightly confused about Fig. 10b. Is temperature kept constant (similarly to snowfall being constant in Fig. 10a)? Please clarify in the caption.

19) L279-280: The increases in the Tarim basin are just on the very edge of the basin.

Can you confirm that the specific grid points that exhibit increases in moisture contributions have undergone an increase in irrigation?

20) L280: You say that the contribution is mainly in May to July, but only May is shown in Fig. 12.

21) L306-307: Do you mean the correlation is weaker because surface fluxes are lower in winter?

22) L335: After "Once the groundwater is depleted, the glaciers in WKSK will also receive less snowfall from this region", you should insert, "according to our results", or something similar.

Technical corrections

1) L122: if → of

2) L126: "less than 1%" should be "more than 99%", unless I misunderstand?

3) L147: rare → sparse

4) L153: of → from

5) L185: extremes → maxima

6) Figure 5 caption: insert "annual" before snowfall. Same in other figures.

7) L214: think → thin

8) L215: northwestern → southwestern. Same on L295.

9) L229: Fig. 3a → Fig. 8a

---

## Editor Comment (EC1) · Thomas Mölg (Editor) · 20 Dec 2019

Dear authors,

Two referee reports are now available, and I would like to thank Dieter Scherer and the second referee for their time and constructive comments. These provide a great starting point for the revision.

In my access review I noted some potential issues, which I would like you to consider as well.

(A) The model evaluation is not convincing – not due to the lack of observations (which is unavoidable), but because of the statistical methods used. One referee emphasizes the same problem. Please extend the model evaluation by metrics that quantify abso-

lute differences (correlations only tell you about variabilities) and be more systematic (c.f., one referee is surprised that annual/May-Sep/and July precipitation are selected). The referee suggestions are very helpful in this regard. In the end, you should demonstrate to the reader that there is confidence in the model results and that they are not a mere model product but represent the real world.

(B) The study is mostly descriptive, and the model output is hardly analysed in terms of processes that could explain the glacier and temperature/precipitation patterns. This approach also results from the fact that descriptions are often (too) short (as the paper was obviously compiled for a short-format journal in the beginning), which leaves some things unclear. Any efforts to expand in this respect would surely be appreciated by the future readers.

Minor (specific) remarks are as follows.

26: "but this alone cannot explain ..." // any reference to support the statement?

63: Please justify why only the upper 35 levels are chosen for nudging

65: How do the nudging parameters compare to the standard values suggested by the developers?

Section 2.1 and rest of paper: I am not sure that calling 20x20 km resolution "high" is still appropriate. It was fine some years ago (and I did so too), but in the meantime with growing computational power, km-scale runs over more than a decade are already available.

Section 2.2: Please clarify the ice dynamics part of the model. It is hard to understand from the current descriptions.

103: why are these three variables chosen for the clustering? I can't comprehend why it is a mix of surface variables and pressure-based variables.

106: please say something like "(indicated later in Fig. 11)", otherwise it seems odd
that Figure 11 comes after Figure 1.

135-139: Is the conclusion correct? Even if one data set shows lower absolute values, it doesn't necessarily mean that a trend must also be lower. Please clarify.

Section 3: Do your spatial patterns have any resemblance with those expected from strong westerlies influence as presented by Mölg et al. (2017, JGR Atmospheres, 122, 3875–3891)? I am not raising this point because I am an author of that study, but because that study has a clear relevance with regard to the scientific content of your paper (westerlies should have an impact in the northwest of HMA).

175: Suggest "variable" instead of "parameter"

255: Many glaciers switch from red to blue; is this really a "minor effect"?

291: The description implies that -0.4 or -0.6 should also be white, which is not the case. Please correct the caption.

---

## Author Comment (AC2) · 23 Jan 2020

We thank Prof. Scherer for his very useful comments. We have addressed his comments below. We show how the text in the manuscript has changed, by indicating new text in boldface.

**Comment:** First of all, the entire study depends on the accuracy of downscaled precipitation. It would therefore be of utmost interest to better understand the uncertainties in the WRF output. As the authors correctly state, in-situ meteorological observations are scarce, and there is almost complete lack of data in the WKSK ranges, which makes it difficult to compare the WRF output with independent observations. This is especially true for high altitudes, i.e., the glacierized areas, where observational data are not available. Nevertheless, there are gridded data sets that could be used for comparison. Although they do not cover the entire study period (so far) and thus cannot substitute the ERA-Interim data used for downscaling, they could anyway be compared with the WRF results for shorter periods (as the authors have done with GLEAM data). The new ERA5 reanalysis and the High Asia Refined analysis (HAR) data set (Maussion et al., 2014) are suitable data sets in this respect. ERA5 data, especially the newest ERA5 land data set (https://cds.climate.copernicus.eu/cdsapp#!/dataset/reanalysis-era5land?tab=overview), and the HAR data set do have very high spatial and temporal resolutions, such that they resolve mesoscale atmospheric processes, and thus orographically induced precipitation. HAR data are freely available at www.klima.tuberlin.de/HAR. I would ask to authors to include a comparison of WRF output with these gridded data sets in the article. This could be put into a supplement with only a short paragraph in the main text.

**Reply:** We agree that such a comparison between different datasets will be a great addition to the manuscript, although we note that this does not necessarily increase the confidence in the results in WKSK, given the lack of ground truth for all these datasets.

We now added several paragraphs and two figures in the main text to deal with the comparison:

We also compare our WRF simulations with three similar data products with relatively high spatial resolutions, which have recently become available. We do note that all these datasets suffer from the lack of ground truth in WKSK, which means we cannot determine which dataset performs best in this region.

ERA5 is the follow-up of ERA-Interim (Copernicus Climate Change Service, 2017), with an improved spatial resolution of 0.25°, an improved temporal resolution, a more appropriate model input for e.g. sea surface temperatures, and more assimilated data. ERA5-Land is atmospherically forced by ERA5, and provides an even higher spatial resolution (0.1°) for land surface properties (Copernicus Climate Change Service (C3S), 2019). Finally, we include the HAR dataset with a resolution of 10 x 10 km, which uses WRF to downscale the NCEP FNR reanalysis dataset and re-initialises every day (Maussion et al., 2014). We compare temperatures between May-September, and annual precipitation, which give an indication of the parameters that are most relevant for glacier mass balance modelling. Because of the limited time overlap between the different datasets, we could only fully compare the period 2001-2010. We binned all data to the same 0.5° x 0.5° grid to allow direct comparison. The mean values, trends, and interannual variability are compared in Figs. 3 and 4. It shows that ERA5 and ERA5-Land are nearly identical, and we only refer to ERA5 below. Our WRF model yields a warmer Karakoram than the other three datasets. Generally, the mean temperature differences are relatively minor, except for a warmer Tarim basin compared to HAR. We find very similar temperature trends as ERA5, although with smaller magnitudes. The magnitudes of the trends are also generally smaller than those in the station data (Fig. 2). The WRF interannual temperature variations correlate very well with ERA5, except two areas in the Tarim and the inner Tibetan Plateau. This is not surprising, given that our WRF model is forced by the similar ERA-Interim data. The whole western part of HMA, including WKSK, is especially well-correlated to ERA5. In that region, the correlation with HAR is weaker, but the correlation between HAR and our WRF data is very strong in East HMA. The differences with HAR might be explained by the different forcing, or by the difference in used physics modules, but this requires further study.

Differences between datasets are larger for precipitation, at least for the mean values and interannual variability. Our WRF simulations give results that are relatively wet in the Karakoram, and relatively dry in the Himalaya. However, the precipitation trends are very similar to ERA5 in both pattern and magnitude. An exception is the arid Tarim basin, which has an increasing trend in WRF, but a decreasing trend in ERA5. HAR shows a positive precipitation trend in most of HMA, with a very high trend in the Tarim basin. The correlation of the interannual variability is low in WKSK and parts of Tien Shan, which could be explained by the relatively high influence of the irrigated areas in the Tarim basin on the annual precipitation (de Kok et al., 2018, Fig. 3). Since our WRF model outcome is the only one of the four datasets that explicitly includes irrigation, this could explain the difference in annual variability.

---

## Author Response (AR1)

**Reply to Reviewer 1**

We thank Prof. Scherer for his very useful comments. We have addressed his comments below. We show how the text in the manuscript has changed, by indicating new text in boldface.

**Comment:** First of all, the entire study depends on the accuracy of downscaled precipitation. It would therefore be of utmost interest to better understand the uncertainties in the WRF output. As the authors correctly state, in-situ meteorological observations are scarce, and there is almost complete lack of data in the WKSK ranges, which makes it difficult to compare the WRF output with independent observations. This is especially true for high altitudes, i.e., the glacierized areas, where observational data are not available. Nevertheless, there are gridded data sets that could be used for comparison. Although they do not cover the entire study period (so far) and thus cannot substitute the ERA-Interim data used for downscaling, they could anyway be compared with the WRF results for shorter periods (as the authors have done with GLEAM data). The new ERA5 reanalysis and the High Asia Refined analysis (HAR) data set (Maussion et al., 2014) are suitable data sets in this respect. ERA5 data, especially the newest ERA5 land data set (https://cds.climate.copernicus.eu/cdsapp#!/dataset/reanalysis-era5-land?tab=overview), and the HAR data set do have very high spatial and temporal resolutions, such that they resolve mesoscale atmospheric processes, and thus orographically induced precipitation. HAR data are freely available at www.klima.tu-berlin.de/HAR. I would ask to authors to include a comparison of WRF output with these gridded data sets in the article. This could be put into a supplement with only a short paragraph in the main text.

**Reply:** We agree that such a comparison between different datasets will be a great addition to the manuscript, although we note that this does not necessarily increase the confidence in the results in WKSK, given the lack of ground truth for all these datasets.

We now added several paragraphs and two figures in the main text to deal with the comparison:

[revised manuscript text omitted]

In the discussion, we add: "**Our snowfall trends between 1980-2010 show some similarities, but also major differences with respect to a similar WRF study that did not include irrigation and used another re-analysis dataset (Norris et al.,**

**2018). For instance, our temperature trends do not exhibit the strong summer cooling at low altitudes (e.g. the Tarim basin), and are more in line with station data (Waqas & Athar, 2018; Xu, Liu, Fu, & Chen, 2010) in that respect. However, contrasting precipitation trends in WKSK and southwestern HMA, similar to Fig. 5** *[now 6],* **are also present**
**in ERA5 data and the Norris et al. study (see Farinotti et al. 2020). Although the interannual variability of temperature and precipitation is reasonably reproduced, and our precipitation trends are similar to those in other datasets, our model results are associated with uncertainties, which are partly irreconcilable due to a lack of** *in situ* **measurements in WKSK."**

**Comment:** The authors shall not only provide Pearson correlation coefficients but also further metrics like mean biases, r.m.s. deviations, regression slopes, etc., when comparing their WRF results with those from GHCN stations. I am not convinced that it is necessary to exclude so many GHCN stations by requesting at least 20 year of data coverage. This could be relaxed, or further comparisons may be added. I am also not convinced that it is sufficient to present results only for aggregated time periods, i.e., for annual mean air
temperatures, May-September air temperatures, and July precipitation. Depending on the details of forcing the glacier model by WRF output, more detailed analyses of the WRF uncertainties are required, since snow- and ice melt can be rather variable from year to year, although years might have shown similar mean seasonal values for air temperature and precipitation.

In this respect, I would ask the authors to add more details on the WRF output and its application for forcing the glacier model simulations and the moisture tracking algorithm. In particular, I would like to know the output time step (one hour?).

**Reply:** It is true that the melt can be different per year. However, the glacier mass balance model does not
include these subtleties. It requires a yearly input of temperature and snowfall and shifts the mass balance gradient accordingly to obtain an annual mass balance. In that sense, presentation of mean melt-season temperatures and annual mean precipitation is a reasonable representation of the data used in the glacier mass balance model. We already stated: "To modulate the mass balance gradient of the glacier over time, we applied annual precipitation changes derived from annual changes in WRF snowfall and temperature changes determined from annual changes
in WRF melt season temperatures, i.e. when average daily temperature is above -5 °C. " We add a more detailed description of the glacier mass balance model as follows:

**" To assess the response of the glaciers to the atmospheric forcing, we employ a glacier mass balance gradient model (Kraaijenbrink, Bierkens, Lutz, & Immerzeel, 2017). The model assumes a calibrated mass balance gradient along the glacier, and parameterises downslope mass flux in a lumped procedure that is based on vertical integration of Glen's flow law (Marshall et al., 2011). It also includes a parameterisation for the effects of supraglacial debris on surface mass balance (Kraaijenbrink et al., 2017), i.e. enhancing melt in the case of a shallow debris layer and limiting melt for thicker debris (Östrem, 1959). We modelled all individual glaciers in HMA larger than 0.4 km$^2$ (n=33,587) transiently for the period 1980-2010 (Kraaijenbrink et al., 2017). For ease of comparison with published observations, we select only those larger than 2 km$^2$ for the final analysis, which represent 95% of the glacier volume in HMA. Initial mass balance conditions in 1980 were set to be stable, while all other initial and reference conditions as described in the original study (Kraaijenbrink et al., 2017) were maintained. That is, using ERA-Interim data to locally calibrate the mass balance gradient of each glacier by constraining maximum ablation by a downscaled positive degree day climatology at the glacier terminus, and maximum accumulation by mean annual precipitation over the entire glacier area. The model simulates glacier mass change and evolution using a one-year time step, and hence requires representative annual input of temperature and precipitation. These are used to shift the mass balance curve according to sensitivity of the glacier's equilibrium line altitude to temperature changes, and adapt the maximum accumulation according to changes in precipitation (Kraaijenbrink et al., 2017)."**

For the WRF output, we add: " **Results are output every 6 hours."**

Given, the annual input, we argue that it is then also reasonable to show the comparison with station data for the relevant aggregated temperature and precipitation data. Because of the mentioned difficulties with measuring snowfall accurately, we take the summer period. We now also took the period May-September, drop the 20 mm limit, and lower the number of available years to 15 to include more stations. Especially trends become very uncertain when few years are considered. The melt season used from the WRF output changes per location, but the summer months are likely to be most important. Hence, we compared these for the temperature data of the stations. Before describing the GHCN results, we add:

**"Since the glacier model requires annual input, representation of the interannual variability is especially important. Any constant biases are of less importance, since we use relative interannual variations as input for the glacier model. However, biases in temperature will have an effect on the snow-rain partition."**

Furthermore, we now add trends and biases into a new figure, which replaces Figures 2 and 3, and briefly discuss their results.
We mention the median root-mean-square deviations in the text. We now write:

"We collected meteorological station data from **the Global Historical Climatology Network** (GHCN, Lawrimore et al., 2011, accessed June 2019), and selected those that have at least **15** years of full data between 1980-2010. **To be able to compare the WRF output with the station data, we apply a simple downscaling to the WRF temperatures in the grid that includes**
**the station. We fit a linear temperature lapse rate to the temperatures and grid altitudes of a 2x2° box surrounding the station location. We then correct the WRF temperature by applying the lapse rate to the difference in altitude between the WRF grid and the station. Precipitation can also change significantly with location, but there is no clear relation between precipitation and altitude (Bonekamp et al., 2019; Collier and Immerzeel, 2015). For this simple comparison, we do not apply a downscaling of the WRF precipitation.**

**Our WRF output produces May-September temperatures that are generally higher than the stations in the Tarim basin. However, biases are generally very low on the Tibetan Plateau, with values around 1°C. The median root-mean-square deviation between WRF and the stations is 1.8°C. The stations generally indicate a strong heating trend.** Correlations between the annual variations in annual mean temperatures and mean temperatures between May-September are
given in Fig. 2. They show generally very high correlations, with a lowest value of 0.5 (corresponding to p = 0.005). This implies that the interannual variability is very well reproduced in WRF. **This is despite the fact that many of these stations are situated in urban environments, with a potential heat island effect, a lack of evaporative cooling that is seen for irrigated agriculture, and a very difference surface energy balance than snow-covered areas. Hence, their locations might not be representative of the wider area, which might give rise to biases and trend differences when comparing**
**the stations to the model outcome.**

[Figure]

**Figure 2: Comparisons between 1980-2010 time series of station data and nearest WRF grid for May-September temperatures (a-c) and May-September precipitation (d-f). Columns show temperature bias (a) and precipitation multiplication factor (d), trends (b,e) and Pearson correlation coefficients. The 2000 m-contour is indicated by a solid line**

The stations in Fig. 2 closest to WKSK are almost exclusively in very arid regions, with a significant fraction of snowfall, **which is more difficult to reliably measure than rain (Archer, 1998),** making comparisons of precipitation very uncertain. Fig. 3 shows the **comparison** between time series of May-September precipitation, to limit the effect of snowfall. **Our WRF output is generally wetter than what is measured at the stations, except some locations in the Tarim basin. The median root-mean-square deviation between WRF and the stations is 11.4 mm per month. The stations show that most of the Tarim basin and Tibetan Plateau are seeing an increase in May-September precipitation.** The interannual variations are not represented by WRF as well as they are for temperature, but still show reasonable correlations for most stations, with values around 0.6. "

For the moisture tracking results, we selected the months that had the largest effect on the aggregated snowfall changes, as already stated in the text.

 **Reply to Reviewer 2**

We thank the reviewer for the many useful comments. We have addressed the reviewer's comments below. We show how the text in the manuscript has changed, by indicating new text in boldface.

 **Comment:** 1) In various figures (2,4,8,10,11,12), concise panel labels would be very helpful to allow the reader to immediately see what each panel shows, without having to read the caption.

**Reply:** We agree that this will help readability and have now added concise labels to all these figures, either in the figure or in the colour bar description.

**Comment:** 2) Similarly, in Figs. 6 and 7, a legend would be very helpful so the reader can immediately see what each line represents.

**Reply:** We agree that this will help readability and have now added legends to these figures.

**Comment:** 3) L54-55: By "the amount of irrigation needed to compensate evapotranspiration", do you mean, after subtracting actual precipitation?

**Reply:** This is indeed the case, and we now indicate this as follows: "... compensate evapotranspiration, **after**
 **subtraction of the precipitation,** that..."

**Comment:** 4) L55: PCR-GLOBWB should be defined/spelled out.

**Reply:** We now add: "... that was calculated by the **PCRaster Global Water Balance model** (PCR-GLOBWB; ..."

**Comment:** 5) L71-73: How are these concentrations of the various greenhouse gases fed into the model? Is it through the radiation scheme?

**Reply:** This is indeed hard-coded in the radiation scheme, which we change for every year. We now state: "

Annual concentrations of $CO_2$, $CH_4$, and $N_2O$, **which are manually set in the RRTMG radiation module, ... "**

**Comment:** 6) L76: What is meant by "convergence between months"?

**Reply:** This was to check whether the monthly spin-up caused discontinuities in the time series. We rephrase the sentence as follows: " We checked **whether temperatures and precipitation at the end of a month agreed with those at the end of the spin-up period for the subsequent month** and they agreed within a few percent for all selected points."

**Comment:** 7) L92: What is meant by "both deltas"?

**Reply:** We now rephrase the sentence as follows: "... the reference for **the changes in temperature and precipitation** was taken ..."

**Comment:** 8) L152-154: In this sentence, it sounds like the implicit assumption is that the measurements are biased, but assuming these biases are constant in time, then we can use them to evaluate WRF's interannual variability. This should be made explicit.

**Reply:** It is not so much the problem that we assume a constant bias. Rather, it is the complete lack of data in the places where we are most interested in, meaning we can only compare it to measurements that are relatively far away. We try to make this clearer by stating: " **Although not covering the glacierised areas of interest,** we compared our WRF output with data of the region **surrounding** WKSK, to ensure that the WRF output is a reasonable representation of the regional climate between 1980-2010."

**Comment:** 9) L156: GHCN has not been defined.

**Reply:** We now state: "... from **the Global Historical Climatology Network (GHCN) ...**"

**Comment:** 10) L159-160: Please explain the relevance of many of these stations being situated in urban environments.

**Reply:** We now state: " This implies that the interannual variability is very well reproduced in WRF. **This is despite the fact that many of these stations are situated in urban environments, with a potential heat island effect, a lack of evaporative cooling that is seen for irrigated agriculture, and a very difference surface energy balance than snow-covered areas. Hence, their locations might not be representative of the wider area, which might give rise to biases and trend differences when comparing the stations to the model outcome.**"

**Comment:** 11) L166-167: Implicit in this sentence is that the stations measure snow less reliably than rain. Please make this explicit and provide a reference.

**Reply:** This is indeed the case. We now state: "... with a significant fraction of snowfall, **which is more difficult to reliably measure than rain (Archer, 1998),** making comparisons of precipitation very uncertain."

**Comment:** 12) L185: How do you know that the discrepancy is only in part due to the different spatial resolution? Have you quantified the effect of the spatial resolution?

**Reply:** We averaged over identical large areas to come to this conclusion. We now state this more explicitly: "... **as is evident from e.g. averaging over 1x1° areas**."

**Comment:** 13) Figure 6 is never referenced in a meaningful way. This figure shows nicely that there is no clear distinction between growing and shrinking glaciers in terms of temperature trends, but that there is a clear distinction in terms of snowfall trends. It would be nice to have some words to this effect in the text.

**Reply:** We agree and add: " **Fig. 6 shows that the trend and the interannual variability of temperature are very similar for nearby regions of both growing and shrinking glaciers.** The snowfall trends **in Fig. 5** have a very different pattern, with most of the Tibetan Plateau showing an increase and the western and southern mountain ranges, such as the Himalaya and the Hindu-Kush, showing a decrease in snowfall. **Furthermore, the mean level, the trend, and the interannual variability of snowfall is quite distinct for the two nearby regions of contrasting glacier mass balance trends.** "

**Comment:** 14) Figure 6 caption: how is representativeness of the bins determined?

**Reply:** The representativeness was not checked, but we simply picked two nearlby points with contrasting mass balances. We increased the representativeness by averaging over larger areas, and modify the caption as follows:

**"...for two nearby 2x3° bins that have, on average, growing glaciers (38-40° N, 73-76° E, blue lines) and shrinking glaciers (35-37° N, 72-75° E, orange, dashed lines)."**

**Comment:** 15) L230-232: It is quite confusing when you say "our simulations only go out to 2010, but we
compare our results for 2000-2008". Why not compare results up to 2010? If 2008 is as far as the observations go, then the limitation is in the observations, not the simulations.

**Reply:** The phrasing was indeed confusing. The observations mostly go to periods later than 2010, but the 2000-2008 period was also given in Brun et al. (2017), although it is less accurate. We now rephrase as
follows: " A more detailed quantitative comparison of the above results and the observed mass balances is hampered by the fact that our simulations only go out to 2010, **and hence we cannot compare with the most recent, and most accurate geodetic mass balance data. However,** we compare our results for the **intermediate** period 2000-2008, **as presented by Brun et al. (2017),** in Fig. 9."

**Comment:** 16) L232-233: As well as the model showing too little growth for the growing glaciers, it shows different glaciers growing to the ones in the observations (Fig. 9). Are the growing glaciers in the model and observations at least in the same areas?

**Reply:** They are indeed. We already mentioned: " In fact, all points where we model glacier growth in Fig. 8a also
show growth or stable conditions in observations (Brun et al., 2017; Kääb et al., 2015), except one point in Kääb et al., (2015)." After the comparison in Fig. 9, we add: " **However, in both cases the growing glaciers are only present in the same region, mainly WKSK and the Tibetan Plateau."**

**Comment:** 17) L259: Presumably the low glacier temperature sensitivity in the WKSK is because, even with warming, temperatures in in the WKSK are still generally below freezing? This could be clarified. Or if there is a different reason?

**Reply:** Although such a narrative is sometimes employed, it is not really true that the glaciers in WKSK always experience negative temperatures. A glacier in balance loses as much mass by melt/sublimation as it has gained by snowfall, when averaged over a long period. Because the accumulation zone in WKSK is indeed very high, the glaciers need to extend down to warmer temperatures to be in balance. We now add: " The reduced temperature sensitivity is in line with previous work (Sakai & Fujita, 2017; Wang, Liu, Shangguan, Radic, & Zhang, 2019), **which argue that the generally large masses of the glaciers, and high equilibrium line altitudes, are important in explaining the lower temperature sensitivity in WKSK**. "

**Comment:** 18) I am slightly confused about Fig. 10b. Is temperature kept constant (similarly to snowfall being constant in Fig. 10a)? Please clarify in the caption.

**Reply:** This is indeed the case, and we now add, similar to 10a: " ... and a spatially uniform and constant snowfall increase of +0.5% yr$^{-1}$ of the annual mean value**, with temperature kept constant** (b)."

**Comment:** 19) L279-280: The increases in the Tarim basin are just on the very edge of the basin. Can you confirm that the specific grid points that exhibit increases in moisture contributions have undergone an increase in irrigation?

**Reply:** These are indeed the irrigation areas. In our model, mainly the Yarkand area show largest increase in irrigation, which is very close to the edge of the basin. We now add: "The regions with the second largest increases are the areas in the Tarim basin **where irrigation has increased the most**, ..."

**Comment:** 20) L280: You say that the contribution is mainly in May to July, but only May is shown in Fig. 12.

**Reply:** We show May, because it is the month with the largest contribution to the increase in snowfall. We now clarify as follows: "... which contributes mainly in May-July, **with May showing the largest resulting increase in snowfall** (see Figs. 7 and 12)."

**Comment:** 21) L306-307: Do you mean the correlation is weaker because surface fluxes are lower in winter?

**Reply:** Mainly the snowfall is less in WKSK in winter. We now clarify as follows: " since this region contributes **relatively more** in winter (Fig. 12)**, when less snowfall reaches WKSK (Fig. 7)**."

**Comment:** 22) L335: After "Once the groundwater is depleted, the glaciers in WKSK will also receive less snowfall from this region", you should insert, "according to our results", or something similar.

**Reply:** We agree, and add the following: " Once the groundwater is depleted, **our results suggest that** the glaciers in WKSK will also receive less snowfall from this region, resulting in their retreat."

**Comment:** Technical corrections

1) L122: if→of
2) L126: "less than 1%" should be "more than 99%", unless I misunderstand?
3) L147: rare→sparse
4) L153: of→from
5) L185: extremes→maxima
6) Figure 5 caption: insert "annual" before snowfall. Same in other figures.
7) L214: think→thin
8) L215: northwestern→southwestern. Same on L295.
9) L229: Fig. 3a→Fig. 8a

**Reply:** We corrected all these technical issues as suggested by the reviewer, except point 3), which was correct (we talk about validation of our model, not validation from our model). We thank the reviewer for the very careful reading.

**Reply to the editor**

We thank the editor for also reading the paper very carefully and giving very useful feedback. We have addressed the editor's comments below. We show how the text in the manuscript has changed, by indicating new text in boldface.

**Comment:** (A) The model evaluation is not convincing – not due to the lack of observations (which is
unavoidable), but because of the statistical methods used. One referee emphasizes the same problem. Please extend the model evaluation by metrics that quantify absolute differences (correlations only tell you about variabilities) and be more systematic (c.f., one referee is surprised that annual/May-Sep/and July precipitation are selected). The referee suggestions are very helpful in this regard. In the end, you should demonstrate to the reader that there is confidence in the model results and that they are not a mere model product but represent
the real world.

**Reply:** Although comparing different datasets is certainly a useful exercise, convergence among datasets is not necessarily is a good measure of reliability or confidence in the model, if there is a lack of ground truth measurements. There is no good way of telling which dataset is closer to the truth in most of WKSK. However,
we now add a much more detailed comparison with station data and other datasets, including 3 new figures. Please see the reply to Referee 1 for the implementation.

**Comment:** (B) The study is mostly descriptive, and the model output is hardly analysed in terms of processes that could explain the glacier and temperature/precipitation patterns. This approach also results from the fact
that descriptions are often (too) short (as the paper was obviously compiled for a short-format journal in the beginning), which leaves some things unclear. Any efforts to expand in this respect would surely be appreciated by the future readers.

**Reply:** We partly disagree with this assessment. We explicitly used the moisture tracking algorithm to directly
analyse the processes that could explain the differences in precipitation, which we show to be more important than differences in temperature. This direct approach means that there is less need for causes of the changes in precipitation to be demonstrated indirectly or circumstantially, e.g. by showing figures of wind fields, evaporation trends, etc., as is commonly done.

We do agree that our descriptions are fairly concise. We did expand our paper by a fair amount before submitting to The Cryosphere (note that we have already included 12 figures in the manuscript), and we tried to be concise, yet complete. We now addressed all the specific issues that caused confusion by the referees, added more description of the glacier model, and we add more discussion on the role of the westerlies (see below).

**Comment:** 26: "but this alone cannot explain..." // any reference to support the statement?

**Reply:** We assumed the reader knows that the sensitivity of a system only influences the rate of response to a disturbance, and not the sign of the response. We try to clarify this by adding: "... but this alone cannot explain
why some glaciers are actually growing, **since either a decrease of ablation or an increase in accumulation is needed**."

**Comment:** 63: Please justify why only the upper 35 levels are chosen for nudging

**Reply:** We add the following sentence: " **This ensures the large-scale upper-atmospheric circulation closely follows
that of ERA-Interim, whereas near the surface, the model is more determined by the physics in WRF, e.g. evaporation in irrigated areas.** The **nudged levels and the** values of the nudging parameters have been found to perform well in similar studies ...**"**

**Comment:** 65: How do the nudging parameters compare to the standard values suggested by the developers?

**Reply:** We add the following sentence: " **The default values for all three parameters are 0.0003 s$^{-1}$ in WRF.**"

**Comment:** Section 2.1 and rest of paper: I am not sure that calling 20x20 km resolution "high" is still appropriate. It was fine some years ago (and I did so too), but in the meantime with growing computational
power, km-scale runs over more than a decade are already available.

**Reply:** We agree. When we started work on this paper, full ERA5 data was not yet available, but things have indeed moved fast. We now rephrase the conflicting sentence in Section 2.1 as follows: **"...** to obtain a climate dataset between 1980-2010 for High Mountain Asia **with a resolution higher than that of ERA-Interim**." We could not find other instances where we claim the resolution to be high.

**Comment:** Section 2.2: Please clarify the ice dynamics part of the model. It is hard to understand from the current descriptions.

**Reply:** We now add: " **The model assumes a calibrated mass balance gradient along the glacier, and parameterises downslope mass flux in a lumped procedure that is based on vertical integration of Glen's flow law (Marshall et al., 2011)."**

**Comment:** 103: why are these three variables chosen for the clustering? I can't comprehend why it is a mix of surface variables and pressure-based variables.

**Reply:** This variable mix was chosen, because they are relevant for the glaciers on the surface, and for the moisture transport higher in the atmosphere. We now add: " **In this way, we delineate regions that have similar surface variables, relevant for the glaciers. Furthermore, these regions are also under the influence of similar winds, relevant for the moisture transport."**

**Comment:** 106: please say something like "(indicated later in Fig. 11)", otherwise it seems odd that Figure 11 comes after Figure 1.

**Reply:** We agree and add the "**later**".

**Comment:** 135-139: Is the conclusion correct? Even if one data set shows lower absolute values, it doesn't necessarily mean that a trend must also be lower. Please clarify.

**Reply:** This is indeed only the case when all values are scaled in the same way. We now add: **"...** the absolute values of the trends will be lower in the WRF domain than outside **if there is a scaling factor in moisture flux between the**

**two datasets**. The trends in the Tarim basin will **then** be underestimated with respect to regions such as the Caspian Sea and the Junggar basin."

**Comment:** Section 3: Do your spatial patterns have any resemblance with those expected from strong westerlies influence as presented by Mölg et al. (2017, JGR Atmospheres, 122,3875–3891)? I am not raising this point because I am an author of that study, but because that study has a clear relevance with regard to the scientific content of your paper (westerlies should have an impact in the northwest of HMA).

**Reply:** This is indeed a very interesting point, and westerlies must indeed be important. This is also clear from Figs. 11 and 12, which show that most of the changes in precipitation in southwestern HMA, which mainly occurs in winter, correspond to source changes west of HMA. The pattern of precipitation trends indeed somewhat matches that expected from an increase in summer westerlies, as described by Mölg et al., but the situation is clearly more complex than simply a change in summer westerlies. We now discuss this issue in a new paragraph in the discussion:

**The pattern of precipitation trends in Fig. 5b roughly matches the pattern that is expected from an increasing influence of summer westerlies, as shown by Mölg et al. (2017). These westerlies are also associated with strong heating and drying trends of the Indus Basin. An increase in irrigation also produces a very similar precipitation pattern, yet causes**

**a cooling and wetting of the Indus Basin (de Kok et al., 2018). Our JAS trends of near-surface temperature and specific humidity (Fig. 13) indicate mostly cooling and wetting trends, which is more in line with the increase in irrigation than with the increase in summer westerlies. ERA5 data for JJA also indicates a similarly strong irrigation effect in the Indus basin (Farinotti et al., 2020). The moisture tracking results (Figs. 11 and 12) indicate that much of the additional snowfall occurs in spring and summer, and originates from the East, with a large role for the irrigated areas. However,**

**the May westerlies clearly have an important role in transporting the increase in evaporation from the Caspian Sea (Chen et al., 2017) to WKSK.  Besides the Caspian Sea, the westerlies are mainly associated with a decrease in snowfall when the whole year is considered (Fig. 11a). The pattern of precipitation trends in Fig. 5b is not only the result of changes in summer. The decrease in precipitation in southwestern HMA is also clearly associated with westerly winds in winter, but not those in summer (see Figs. 7d and 11c).**

[Figure]

**Figure 13: Trends between 1980-2010 of near-surface temperature (a) and specific humidity (b) between July-September, averaged over 0.5x0.5° bins for clarity. The 2000 m elevation contour is indicated by a solid line.**

**Comment:** 175: Suggest "variable" instead of "parameter"

**Reply:** We changed the text as suggested.

**Comment:** 255: Many glaciers switch from red to blue; is this really a "minor effect"?

**Reply:** The figure shows that the effect of precipitation is smaller than the effect of temperature. We try to clarify the sentence as follows: "... temperature, **while the decrease in precipitation gives a much smaller mass balance response in this region**.

**Comment:** 291: The description implies that -0.4 or -0.6 should also be white, which is not the case. Please correct the caption

**Reply:** We add "absolute" before "magnitude".

[revised manuscript text omitted]

---

## Referee Report (RR1)

Employing a simple glacier mass balance model forced with dynamically refined ERA-Interim at 20km with irrigation, the study mainly focuses on reproducing the observed mass balance gradients of the WKSK glaciers, as the role of irrigation on summer snowfall increase as well as the impact of increasing westerly solid precipitation both resulting in positive or balanced mass balances have already been reported either by the authors or other studies. I very much like the idea of the study.

Quite common among the most of climate modelling studies focusing the WKSK region is the lack of their validation against (high-altitude representative) observations within the WKSK region itself, and especially against those that actually include snow, although such observations are quite a few.

Lacking such validation initially, revised study now includes more stations, adjust WRF-20km temperatures prior to their comparison with the stations, and also, validates the WRF-20km against the ERA5, ERA5-Land and HAR fine scaled reanalysis datasets.

Besides above additions and revisions in the study, I am of the view that although observed mass balances of the WKSK glaciers are reproduced - at least to some degree of satisfaction - however path to such outcome seriously lacks realism. There are large and unquantified uncertainties at each step of reproducing the delicate mass balance gradient of the WKSK glaciers, which need to be first reduced to the extent possible, quantified and then assessed for their possible impacts on the results and conclusions. My particular concerns are given below:

**MAJOR CONCERNS:**

To establish the robustness of the WRF-20km simulations for further use in mass balance modelling of the WKSK glaciers, its scale should be refined and its validation must be performed over the complex terrain of the focused regions against few available high-altitude observations from the WKSK region, instead of only against the stations from non-glaciated surrounding areas. Because, relative to the less complex terrain of the surrounding areas, coarse grid size dynamical refinement (as in the study) perform poor over highly concentrated topography of the western Tibetan Plateau, Karakoram and adjacent regions, irrespective of the forcing datasets, featuring substantial cold (6-10deg) and wet biases. These biases are spatially heterogeneous and are difficult to adjust statistically, particularly for precipitation. Not surprising that in the revised study, the WRF-20km temperatures have been adjusted only for comparison against the station observations but not for calculating melt season temperatures for mass balance modelling, whereas, precipitation was not adjusted at all. Substantial cold biases at 20km grid size over complex terrain may result in overall shorter melt season, reduced energy available for melt and anomalous snowfall amounts. Reducing these biases and then quantifying the effect of remaining biases on the results are therefore fundamental to establish the robustness of climatic simulation, and in turn, of modelling delicate mass balance gradients of the WKSK glaciers. Establishing the robustness of climatic simulation requires at least introducing a convection-permitting scale and resolving valley scale physical processes explicitly, by introducing a further model nest spanning over the WKSK region and then extensively validating it against available high-altitude stations from SIHP WAPDA, PMD, Agha Khan Agency for Habitat, CMA, EVK2-CNR, and others from the WKSK region at least within the 2000-2010 period, which has been used for mass balance comparison.

Further, precipitation in the only used model realization is highly sensitive to the chosen cumulus parameterization, microphysics and related physics in the climate model. Any change in physics leads to significantly different magnitudes of precipitation and signs of change, beside other climatic conditions. Effort to quantify the sensitivity of results to chosen physics in the model is missing. Recommendation from the literature could have been useful too. Chosen model physics is actually based on de Kok et al., 2018, who state that their 7-year simulation was not aimed at accurately reproducing the reality but only to show the effect of irrigation, unlike this study.

The implementation of irrigation in 20 km WRF simulation lacks realism as it perturbs precipitation at each timestep based on monthly crop water demand rate calculated by PCR_GLOBWB forced by different dataset (may be CRU TS2.1 and ERA-40), and most importantly, ignoring the on-ground facts, such as, deficit conditions and the irrigation efficiency. This can lead to anomalously higher moisture availability that yields increased snowfall in the neighboring regions and subsequently can positively affects the mass balance results. Hence, it is important to realistically implement the irrigation in the model to avoid introducing spurious atmospheric moisture amounts that are favorable to the conclusions of the study. Qualitative validation against non-validated proxy evapotranspiration observations does not add to the robustness of the WRF-20km irrigation and requires to be replaced with quantitative validation against reliable datasets including quantification of their uncertainty and subsequent impacts on the conclusions.

**Other comments:**

- What is the statistical significance of presented trends in temperature and snow fall? Do presented slopes actually carry any physical meanings?
- Line 180-185: how it is established that GLEAM provides unrealistically low evapotranspiration in heavily irrigated arid regions in July. Validation of WRF-20km against the employed evapotranspiration proxy observations is completely subjective and unreliable. In fact, landing in the middle of multiple non-validated unreliable datasets does not establish robustness of the WRF-20km simulations.
- I think, unlike west Kunlun Shan, Karakoram region feature accumulation during winter and spring. Validation of winter and spring climatic conditions seems important here.
- Is the water demand calculated based on ERA-Interim? I guess that the water demand calculated by the PCR-GLOBWB was based on CRU/ERA-40 datasets, which are different than those used here. If yes, any explanation on the effects on results should be added.
- How precipitation at each time step was perturbed in the model is not clear. How it has been achieved?
- Implementing irrigation through continuous light rain in the study completely ignores the significant impact of irrigation timing on the climate and seasonality and the state of vegetation. I think direct perturbation of soil moisture is a better approach that imitates irrigation via a flooding of the surface and disregards other reservoirs such as the canopy layer. Hence, it is important to know what effect introducing of continuous precipitation had on realistically reproducing surface parameters? I hope gridded observations at least over the plain areas of south Asia are representative and can be used for validation.
- Describe negative trend of regional glacier mass balance for the WKSK in Figure 7(d).

- Line 156: Within upper Indus basin, observations from a number of automated weather stations are available from SIHP, Pakistan since 1994/1995 up to 4440 m asl and from the long-term PMD stations up to 2200 masl since 1960s or earlier. For example in Norris et al., 2018. Importantly, SIHP automated weather stations include both snow and rain. Additionally, snow heights and snow fall amounts from the Weather Monitoring Posts from the Agha Khan Agency for Habitat and a few observations from EVK2-CNR are a valuable database for validation. For this, station selection criteria can be further relaxed to the available observations as the validation of the whole length of simulation does not seem to be mandatory in the data scarce region.
- Lines 220-225: Unlike Waqas and Athar (2018), several studies suggest statistically significant strong cooling at least in July and September months over both low and high-altitude stations within the HKH region.
- The study mainly focuses on reproducing the annual mass balances of the WKSK glaciers featuring delicate changes using a highly simplified lumped mass balance model. It would have been better to model the mass balance of these glaciers using more sophisticated model and on an intra-annual scale as measurements/variables from the model are not an issue.

---

## Editor Decision (ED1)

Dear authors,

Thank you for revising the manuscript and for your responses. Since one point of critique in the first round (open discussion phase) was serious ("insufficient model validation") and the revised MS has changed quite substantially, I decided to ask a third, new referee for an independent opinion. This person has a strong background in regional atmospheric modeling. You will find his/her review attached, and you will see that the referee has major concerns with the study and rated the scientific quality of the paper as "poor".

This referee points out a "lack of realism", which links directly to the aforementioned "insufficient model validation", suggesting that the paper has not improved convincingly in that respect. In particular, the uncertainty in precipitation is emphasized by the new referee, which also connects to Dieter Scherer's comment "...the entire study depends on the accuracy of downscaled precipitation. It would therefore be of utmost interest to better understand the uncertainties in the WRF output." from the first round of review. In addition, my own review (see attached) raises a question in the same direction (are model and observations in agreement? See point **(2)** of my comments). Therefore, revisions are necessary, and I will make a final decision after the re-submission and your responses whether the manuscript can be considered for publication or not.

I hope these comments are helpful for clarifying your study further.

Thomas Mölg
Handling Editor &
Co-Editor-In-Chief TC

**Editor Review of revised MS**

My own comments on the revised version concern two areas.

**(1)** Readability
Below I list ~20 examples that I caught during my reading, which illustrate a lack of precision in statements or procedures. While each case on its own is probably a minor problem, all together make reading the paper quite hard. Please go through the paper carefully and revise these and similar problems to enhance the unambiguity of statements. I hope you can see that the below examples are not reader-friendly.

96-100: Several instances of "temperature"; it is unclear whether they refer to air or the glacier (surface).

101-105: "snow" and "snowfall" are used here; do they refer to the same (I guess you mean snowfall as you refer to solid precipitation)?

123: "amount of water"; all phases or just a particular one?

181: RMSE of 1.8 °C; were annual, seasonal, monthly values used in the calculation? It is not obvious.

Figure 2 caption: What is the "nearest WRF grid" in this context? Do you mean "nearest grid point"?

Figure 2: The figure is supposed to show a station/WRF comparison. How do panels (b) and (e) fit here? They show a station trend.

198: RMSE of 11.4 mm/month; were annual, seasonal, monthly values used in the calculation? Again, it is not obvious (per month can also be used as unit for mean seasonal or annual values).

Figure 2: it is referred to a lot in the text, but almost always without specification of which panel is meant (a few times I understand that the entire figure is addressed, but that can't be the case always).

232: is "figure 3 in de Kok et al., 2018" meant? Figure 3 in the present paper is a temperature figure.

Figure 3: Time step for the correlation calculation not clearly specified (annual, season, …).

Figure 4: Time step for the correlation calculation not clearly specified (annual, season, …).

279-281: How can changes (increase/decrease) be inferred from Figure 8, which shows the mean diurnal cycles?

Figure 7: The captions says regions with "growing" and "shrinking" glaciers, but the legend shows WKSK and SW HMA. Are these two definitions exactly the same regions?

Figure 9: Is the scale bar adjusted to the min/max values in the maps? I can't see much dark blue in the maps.

332: "snow" or "snowfall"; I guess you mean the latter?

382-385: Same as above, which time step is used for calculating the correlations? "Interannual" can also compare seasons or months between years (or annual values).

Discussion starting in 409: "roughly matches" is not clear enough. Also, your Figure 6b shows snowfall trends, yet you refer to precipitation here. The cited reference also shows precipitation. Please be consistent for comparisons.

General: What tests are used to determine the p values?

General: Are the trends tested for significance?

General: with the addition of new data sets, mixing up Methods and Results has become more serious than in the first version. The readability would benefit from having more descriptions of data and technical procedures before the results section (Section 3).

[Figure]

**(2)** Scientific Contents
My main topical comment refers to Section 3.3, where one key message is that areas of growing or shrinking glaciers are consistent in model and observations. While this would be a nice result, I assume that readers will have trouble understanding it when they look at Figure 10. In particular, where the model region tends to be positive (marked Box 1 below) the Brun values are mostly negative, and where the model region tends to be negative (Box 2) the Brun results tend to be positive. One could also conclude that model and observation show the opposite with regard to neutral/stable mass balances. This discrepancy adds to the referee assessment of "a lack of realism".

**Minor**

87: represent 95% → needs a reference

158: "almost complete" is redundant

170: extracted or downloaded instead of "collected"

249: "our model" → which one is meant? The trajectory model?

---

## Author Response (AR2)

We thank the editor for providing more comments to improve our manuscript. Below are our detailed replies to these comments.

**Comment:** Thank you for revising the manuscript and for your responses. Since one point of critique in the first round (open discussion phase) was serious ("insufficient model validation") and the revised MS has changed quite substantially, I decided to ask a third, new referee for an independent opinion. This person has a strong background in regional atmospheric modeling. You will find his/her review attached, and you will see that the referee has major concerns with the study and rated the scientific quality of the paper as "poor".

This referee points out a "lack of realism", which links directly to the aforementioned "insufficient model validation", suggesting that the paper has not improved convincingly in that respect. In particular, the uncertainty in precipitation is emphasized by the new referee, which also connects to Dieter Scherer's comment "...the entire study depends on the accuracy of downscaled precipitation. It would therefore be of utmost interest to better understand the uncertainties in the WRF output." from the first round of review. In addition, my own review (see attached) raises a question in the same direction (are model and observations in agreement? See point \*\*(2)\*\* of my comments). Therefore, revisions are necessary, and I will make a final decision after the re-submission and your responses whether the manuscript can be considered for publication or not.

**Reply:** We agree that there are unquantified uncertainties in our model, which we already acknowledge in our manuscript. However, many uncertainties are simply unquantifiable due to the lack of data in WKSK, as we also indicate in our manuscript. This is a problem that is relevant for all studies in this region, including those already published. We disagree that our modelling lacks realism, since all our models are ultimately driven by reanalysis data, which are all largely influenced by observations of the surface and atmosphere. However, we now also added further validation with remote sensing data, showing that the atmospheric moisture in our model excellently reproduces observations, much in contrast to the statements of the referee. Using a combination of techniques, our study is the first to reasonably reproduce the pattern of mass balances in HMA and we provide many new insights into what might cause this anomalous mass balances. Furthermore, our paper gives useful new avenues for future research. In summary, we think our paper is very much suitable for publication in The Cryosphere. Although our title is a concise summary of our results, we now tried to better acknowledge the uncertainties in our modelling, by changing the title as follows: Towards understanding the pattern of glacier mass balances in High Mountain Asia using regional climatic modelling. We also add to the abstract: "...we reproduce the observed patterns of glacier mass balance in High Mountain Asia of the last decades within uncertainties." The rest of the abstract already has cautious language.

To gain confidence in our modelling results we have performed comparisons and validation efforts that are well beyond what is normally done in similar studies in the published literature. We have:

- Compared our model results with station data, which is what is commonly done, but is challenging due to the scale difference and large uncertainties in precipitation observations in mountains.
- 2) Compared our model results with several reanalysis datasets, which is commonly done in separate papers.
- 3) Compared different ET datasets, which is normally not done and is normally extended to a study in itself.
- 4) Compared our model results with AIRS+AMSU retrievals, which provides a new kind of validation that is not commonly done.

All of our comparisons show that our model compares at least reasonably well, and often very well, with observations for all of the most relevant parameters, and hence certainly does not "lack realism". We also already acknowledge and discuss our model uncertainties to a larger extent than most atmospheric modelling papers in the region, and even globally.

However, the statement about the lack of realism possibly does not only relate to our model validation, but with a misinterpretation of our methods by this referee. We understand there would certainly be a lack of realism if we would have used the WRF fields directly to model the mass balance, or the mass balance gradient, which is what the referee might have assumed. We now provide a detailed discussion of our approach of using relative changes in temperature and precipitation and now demonstrate that our method is much more robust against the potentially large biases in the WRF model (e.g. by the reduced resolution or choice of physics modules), and is hence much more realistic than downscaling the WRF data directly. We also further discuss the uncertainties of our method. We add:

... We show that such a pattern can be reproduced using relative changes in temperature and precipitation in recent decades. Since we used relative changes to force our glacier model, we are less influenced by errors in the absolute precipitation amounts, caused by our low resolution or by our choice of model physics. We illustrate this using WRF runs performed for de Kok et al. (2018) for May-September of two years. We ran WRF at two resolutions: at 20 km with same the physics settings as in this study, but without any nudging, and at 4 km, which is of high enough resolution to explicitly resolve convection and avoid the cumulus parameterisation. There are large local differences in precipitation between the two runs, mainly due to the difference in resolution. However, when the relative ratio of the precipitation is plotted for two years (Fig. 16), similar to what is used in the glacier model, the two set-ups give much more similar patterns. Snowfall gives very similar results, but we decided to show total precipitation, where total numbers and cumulus errors are expected to be even higher. The relative changes in precipitation do not markedly show the topography, in contrast to the individual precipitation fields. Rather, relatively large regions show similar interannual changes in the precipitation. The patterns of precipitation change also agree well between the 20 km results and the 4 km results, despite the very different treatment of the convection and the difference in topographic resolution. The differences between the scaling factor in the two cases can be of the order of tens of percent, which is much smaller than the difference in absolute precipitation amounts that would be needed to model the mass balance directly from the WRF fields. Also temperatures are mutually correlated over larger areas in WKSK (e.g. Forsythe et al., 2017) and the glacier mass balances in HMA also vary mainly over a large scale, suggesting that large-scale weather patterns are on average more important in controlling the interannual variability of temperature and precipitation than the differences between valleys. The use of relative changes in temperature and precipitation has thus made our results more robust against possible errors in the detailed treatment of the complex mountain meteorology.

Figure 16: Precipitation ratios between May-September of two years for the WRF run at 20 km, with cumulus parameterisation (a), that at 4 km resolution, without parameterisation (b), and the two compared, when binned at the resolution of the 20 km run (c). The 2000 m elevation contour is indicated by a solid line.

One of our main sources of error is setting up the initial mass balance gradient, and our assumption that the glaciers are initially in balance. Due to the inertia of the glaciers, the initial condition has relatively large influence on the eventual mass balance decades later, as discussed above. Furthermore, any errors in the mass balance gradient, e.g. due to errors in the downscaling of ERA-Interim data, will affect the temperature and precipitation sensitivities presented here, but will have less impact on the overall pattern of mass balances in HMA, since they are mostly determined by the changes of temperature and precipitation.

**Comment:** Below I list ~20 examples that I caught during my reading, which illustrate a lack of precision in statements or procedures. While each case on its own is probably a minor problem, all together make reading the paper quite hard. Please go through the paper carefully and revise these and similar problems to enhance the unambiguity of statements. I hope you can see that the below examples are not reader-friendly.

**Reply:** Although the editor and all three referees have rated the presentation of the results as "good" or "excellent" in previous rounds, we agree that there is always room for further improvement and we have taken all the points into consideration.

**Comment:** 96-100: Several instances of "temperature"; it is unclear whether they refer to air or the glacier (surface).

**Reply:** We now included "air" before "temperature" six times in this paragraph.

**Comment:** 101-105: "snow" and "snowfall" are used here; do they refer to the same (I guess you mean snowfall as you refer to solid precipitation)?

Reply: We replaced all instance there to say "snowfall".

Comment: 123: "amount of water"; all phases or just a particular one?

Reply: This is indeed all phases of water. We add: "...the total amount of all water in the parcel..."

**Comment:** 181: RMSE of 1.8 °C; were annual, seasonal, monthly values used in the calculation? It is not obvious.

Reply: We add: "...the stations for the time-series of seasonal mean temperatures is 1.8°C."

**Comment:** Figure 2 caption: What is the "nearest WRF grid" in this context? Do you mean "nearest grid point"?

Reply: We now added "point".

**Comment:** Figure 2: The figure is supposed to show a station/WRF comparison. How do panels (b) and (e) fit here? They show a station trend.

**Reply:** The trends give an idea about the content of the station data. We elaborate on this in the text: "The stations generally indicate a strong heating trend, **but also show relatively large differences for close-by stations**." and "**The stations show an increasing trend in May-September precipitation in the western Tarim basin and most of the eastern Tibetan Plateau**."

**Comment:** 198: RMSE of 11.4 mm/month; were annual, seasonal, monthly values used in the calculation? Again, it is not obvious (per month can also be used as unit for mean seasonal or annual values).

Reply: We add: "...for the time-series of seasonal mean precipitation is 11.4 mm per month."

**Comment:** Figure 2: it is referred to a lot in the text, but almost always without specification of which panel is meant (a few times I understand that the entire figure is addressed, but that can't be the case always).

**Reply:** In the text, where the figure is first presented, different panels are discussed, but the panels show very different variables, which we clearly mention in the text. We now also add a reference to the figure panel each time. Besides the text that directly presents Fig. 2, we only find one direct other reference to it in the text. To further clarify, we add the panel number when discussing Fig. 3: "The magnitudes of the trends are also generally smaller than those in the station data (Fig. 2b)."

**Comment:** 232: is "figure 3 in de Kok et al., 2018" meant? Figure 3 in the present paper is a temperature figure.

**Reply:** This is indeed the case. Although the original text is a common format for citing page numbers or figures in other papers, we rephrase now as: "(**Fig. 3 of** de Kok et al., 2018)."

**Comment:** Figure 3: Time step for the correlation calculation not clearly specified (annual, season, ...).

**Reply:** All correlation calculations are performed in a similar way, and we now elaborate in Section 2.4: "Pearson correlation coefficients are calculated **between pairs of different datasets (e.g Figs. 2-5)** using the vectors of annual or seasonal mean values, **with one value for each year. The figures indicate over which period the mean is taken for each year.** The trends shown in Figs. 2-5, 8, and 17 are the slopes from linear fits to **these** vectors.

**Comment:** Figure 4: Time step for the correlation calculation not clearly specified (annual, season, ...).

Reply: All correlation calculations are performed in a similar way, and we now elaborate in Section 2.4: "Pearson correlation coefficients are calculated between pairs of different datasets (e.g. Figs. 2-5) using the vectors of annual or seasonal mean values, with one value for each year. The figures indicate over which period the mean is taken for each year. The trends shown in Figs. 2-5, 8, and 17 are the slopes from linear fits to these vectors. "

**Comment:** 279-281: How can changes (increase/decrease) be inferred from Figure 8, which shows the mean diurnal cycles?

**Reply:** We are confused by the comment of the editor here. Nowhere in the paper do we show the diurnal cycle. We do show the mean seasonal cycle in Fig. 8, but we also show the trends in panel d. We now mention this panel explicitly.

**Comment:** Figure 7: The captions says regions with "growing" and "shrinking" glaciers, but the legend shows WKSK and SW HMA. Are these two definitions exactly the same regions?

**Reply:** The two areas are sub-areas of WKSK and SW HMA, which we now clarify in the caption: "...for two nearby 2x3° bins **in WKSK and southwestern HMA** that have..."

**Comment:** Figure 9: Is the scale bar adjusted to the min/max values in the maps? I can't see much dark blue in the maps.

**Reply:** The positive part of the scale bar follows those of the much-read works of Kääb et al. and Brun et al., which allows for better comparison with these works.

Comment: 332: "snow" or "snowfall"; I guess you mean the latter?

Reply: We now write "snowfall"

**Comment:** 382-385: Same as above, which time step is used for calculating the correlations? "Interannual" can also compare seasons or months between years (or annual values).

Reply: All correlation calculations are performed in a similar way, and we now elaborate in Section 2.4: "Pearson correlation coefficients are calculated between pairs of different datasets (e.g. Figs. 2-5) using the vectors of annual or seasonal mean values, with one value for each year. The figures indicate over which period the mean is taken for each year. The trends shown in Figs. 2-5, 8, and 17 are the slopes from linear fits to these vectors. "

**Comment:** Discussion starting in 409: "roughly matches" is not clear enough. Also, your Figure 6b shows snowfall trends, yet you refer to precipitation here. The cited reference also shows precipitation. Please be consistent for comparisons.

**Reply:** We now attempted to clarify the section, including adding an extra panel, showing the WRF JAS total precipitation trend. We now write:

The pattern of **snowfall** trends in Fig. 8b roughly matches the **precipitation** pattern that is expected from an increasing influence of summer westerlies, as shown by Mölg et al. (2017). **From this similarity, one could wonder whether the snowfall pattern from Fig. 6b is mainly caused by summer westerlies.** These **summer** westerlies are also associated with strong heating and drying trends of the Indus Basin. An increase in irrigation also produces a very similar precipitation pattern **as the pattern for summer westerlies**, yet causes a cooling and wetting of the Indus Basin (de Kok et al., 2018). Our JAS trends of near-surface temperature and specific humidity **from WRF** (Fig. 17) indicate mostly cooling and wetting trends **in the Indus basin**, which is more in line with the increase in irrigation than with the increase in summer westerlies. ERA5 data for JJA also indicates a similarly strong irrigation effect in the Indus basin (Farinotti et al., 2020), **as indicated by a wetting and cooling trend**. The moisture tracking results (Figs. 14 and 15) indicate that much of the additional snowfall occurs in spring and summer, and originates from the East, with a large role for the irrigated areas. The decrease in precipitation in southwestern HMA is also clearly associated with westerly winds in winter, but not those in summer (see Figs. 10d and 14c). The pattern of **snowfall** trends in Fig. 8b is **thus** not only the result of changes in summer. **When only JAS is considered, the pattern of precipitation trends look different from the annual snowfall trends (Fig. 17b). Therefore, the summer westerlies are likely not the main driver for the snowfall pattern seen in Fig. 8b. However, the May westerlies clearly have an important role in transporting the increase in evaporation from the Caspian Sea (Chen et al., 2017) to WKSK. Besides the Caspian Sea, the westerlies are mainly associated with a decrease in snowfall when the whole year is considered (Fig. 14a).**

---

## Author Response (AR3)

Dear Prof. Mölg,

Many thanks for going over the manuscript again. We are happy that the manuscript is now suitable for publication. Below are our responses to your comments, and shows how we edited the manuscript.

**Comment:** Thank you very much for the thorough revision. In my opinion, the content related to two former critique points has substantially improved: (a) showing the robustness of the WRF precipitation output for glacier modeling, and (b) clarifying the method to simulate irrigation effects.

While some discussion points remain (e.g., (i) I do not understand why trends should not be tested for significance; internal variability is one reason for such testing, not an argument against it; (ii) describing data sets in the results instead of methods; (iii) evaluating humidity at ~450 hPa, which is from an air layer that is typically not a dominant factor for the processes of precipitation formation), these do not concern technically incorrect issues.

**Reply:** We agree that these points can be further discussed, and add short comments here. (i) We agree that the internal variability is important, but our point was mainly concerned with the link of significance with physical meaning of the trend, as referred to by the third referee. Since the significance does not influence our eventual mass balance modelling, we have left it out. (ii) We agree that these datasets can be equally well introduced in the methods, but since it is the first time the output of our modelling is used, we have decided to include everything in this one section, to have the information less spread out over the document. (iii) The layer does give a good indicating of the large-scale atmospheric patterns. It would indeed be even better to have information at lower altitudes, but, as we indicate, uncertainties in the data are likely much higher there.

**Comment:** As the manuscript has received extensive feedback to this point, my opinion is that it contains enough valuable ideas to enrich the literature. Hence, I see the paper in a good position for publication. Please consider the following minor (editorial) comments:

37: "directly" twice; suggest deleting one of them.
**Reply:** We agree and have now deleted the first "directly".

**Comment:** 67: Please add unit to all three numbers.
**Reply:** We have now added the same units to all three numbers.

**Comment:** 96-100: If you refer to Marzeion's study with respect to "other models", I would move the citation a little more ahead in this part.
**Reply:** We have now moved the reference to the first part of the sentence.

**Comment:** 192: "also given in Fig. 2".
**Reply:** We changed the text as suggested.

**Comment:** 193: suggest "… variability is reproduced in WRF"; I am not sure that r values between 0.5 and 0.7 (i.e. explaining less than half of the variance") can be called "very well". But I leave it to the authors.
**Reply:** We now removed the "very".

**Comment:** Figure 3+4 captions: Please add the time period of the comparison (2001-2010, right?).
**Reply:** Indeed, we added "between 2001-2010".

**Comment:** 455: suggest "which is a resolution typically used to explicitly …" (please avoid "high enough" – many atmospheric physicists will say 4 km is not high enough).
**Reply:** Good point, and we change the sentence as suggested.

**Comment:** 463: delete "very"
**Reply:** We deleted the word.

**Comment:** 464: "on the order of"
**Reply:** My UK PhD supervisor always changed my "on the order of" to "of the order of" and this seems indeed a correct way of writing. We therefore leave the statement.

**Comment:** 466: Please clarify "mutually correlated"? Do you refer to spatial autocorrelation?
**Reply:** Indeed, we now state: "…show a spatial autocorrelation…"

**Comment:** 474: delete first word "that"
**Reply:** We replaced the word "that" to read: .."… run at 4 km…"

**Comment:** Figure 17 caption: "(b)" appears twice, but it should be "(c)" in one case.
**Reply:** We have replaced the second (b) by (c).

**Comment:** 538: "far too insignificant" – suggest "the lack of significance prevents to draw …" or similar; "far too" is unclear.
**Reply:** We now replaced this part by: "…but a lack of significance prevents us from drawing…"

**Comment:** Last, I would like to make a personal remark. Throughout the review process, I found the slightly aggressive tone in the author responses to criticism somewhat problematic, compared to the many responses I have seen in my 10 years as editor. While there is no doubt that referees and editors are not always right and some of their comments may not be relevant, nobody is raising critique arbitrarily. Multiple author responses left an aftertaste for me along the line "the others are wrong and how dare they". Writing responses is of course a personal style (and maybe taught differently in different places), but I wanted to share my impression for the present case. Objectivity and respect should be strong commitments from editors and referees, but also from authors.

**Reply:** We sincerely regret that our replies were seen as aggressive. We certainly did not aim for such a tone. We only tried to make clear how we disagreed with some of the points of the reviewers, and give detailed explanations of our own reasoning. We tried to do this is a clear, factual, and to-the-point manner, in order to aid the review process. We always tried to look at our manuscript from the point of view of the reviewer, and change the manuscript such that the reviewer's concerns could be alleviated as well as we thought possible. Therefore, we wrote extensive replies and significantly changed our manuscript based on the reviewers' suggestion, which we hope you can see is a sign that we have treated the reviewers' comments with care and respect. We did not make any respectless comments towards the reviewers or editor, to our knowledge. We deeply apologise for the unpleasant experience that our reply has given and we hope future interactions are more pleasant.

[revised manuscript text omitted]